# Curse of Slicing: Why Sliced Mutual Information is a Deceptive Measure of Statistical Dependence

**Alexander Semenenko[1], Ivan Butakov[1,2,4], Ivan Oseledets[1,3,4] & Alexey Frolov[1]**
[1]Applied AI Institute [2]Moscow Independent Research Institute of Artificial Intelligence
[3]AXXX [4]Institute of Numerical Mathematics, RAS
Moscow, Russia
{semenenko, ivan.butakov}@applied-ai.ru

## Abstract

Sliced Mutual Information (SMI) is widely used as a scalable alternative to mutual information for measuring non-linear statistical dependence. Despite its advantages, such as faster convergence, robustness to high dimensionality, and nullification only under statistical independence, we demonstrate that SMI is highly susceptible to data manipulation and exhibits counterintuitive behavior. Through extensive benchmarking and theoretical analysis, we show that SMI saturates easily, fails to detect increases in statistical dependence, prioritizes redundancy over informative content, and in some cases, performs worse than correlation coefficient.

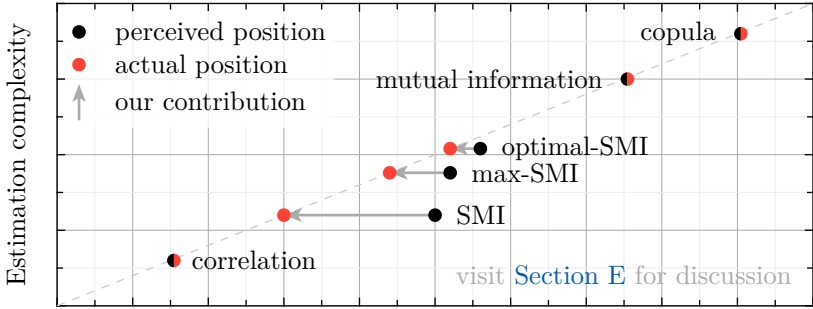

Ability to capture complex statistical dependencies

## 1 Introduction

Mutual information (MI) is a fundamental and invariant measure of nonlinear statistical dependence between two random vectors, defined as the Kullback-Leibler divergence between the joint distribution and the product of marginals (Polyanskiy and Wu, 2024):

$$\mathsf{I}(X;Y) = \mathsf{KL}\big[\mathbb{P}_{X,Y} \,\big\|\, \mathbb{P}_X \otimes \mathbb{P}_Y\big].$$

Due to several outstanding properties, such as nullification only under statistical independence, invariance to invertible transformations, and ability to capture non-linear dependencies, MI is used extensively for theoretical analysis of overfitting (Asadi et al., 2018; Negrea et al., 2019), hypothesis testing (Duong and Nguyen, 2022), feature selection (Battiti, 1994; Vergara and Estévez, 2014), representation learning (Bachman et al., 2019; Butakov et al., 2025; Hjelm et al., 2019; Tschannen et al., 2020; Veličković et al., 2019), and studying the mechanisms behind generalization in deep neural networks (DNNs) (Butakov et al., 2024; Goldfeld et al., 2019; Shwartz-Ziv and Tishby, 2017; Tishby and Zaslavsky, 2015).

In practical scenarios, $\mathbb{P}_{X,Y}$ and $\mathbb{P}_X \otimes \mathbb{P}_Y$ are unknown, requiring MI to be estimated from finite samples. Despite all the aforementioned merits, this reliance on empirical estimates leads to the curse of dimensionality: the sample complexity of MI grows exponentially with the number of dimensions (Goldfeld et al., 2020; McAllester and Stratos, 2020). A common

strategy to mitigate this issue is to use alternative measures of statistical dependence that are more stable in high dimensions. However, such measures usually offer only a fraction of MI capabilities. Therefore, it is crucial to maintain a balance between robustness to the curse of dimensionality and the ability to detect complex dependency structures.

To strike this balance, popular techniques often retain MI as a backbone statistical measure but employ dimensionality reduction before estimation. While some studies explore sophisticated nonlinear compression methods (Butakov et al., 2024; Gowri et al., 2024), others favor more scalable linear projection approaches (Fayad and Ibrahim, 2023; Goldfeld et al., 2022; Goldfeld and Greenewald, 2021; Greenewald et al., 2023; Tsur et al., 2023). Among the latter group, the *Sliced Mutual Information* (SMI) (Goldfeld et al., 2022; Goldfeld and Greenewald, 2021) stands out, leveraging uniform random projections:

$$\mathsf{SI}(X;Y) = \frac{1}{\oint_{\mathbb{S}^{d_x-1}} \mathrm{d}\theta} \frac{1}{\oint_{\mathbb{S}^{d_y-1}} \mathrm{d}\phi} \oint_{\mathbb{S}^{d_x-1}} \oint_{\mathbb{S}^{d_y-1}} \mathsf{I}(\theta^\mathsf{T} X; \phi^\mathsf{T} Y)\,\mathrm{d}\theta\,\mathrm{d}\phi. \tag{1}$$

Uniform slicing allows SMI to maintain some crucial properties of MI (e.g., being zero if and only if $X$ and $Y$ are independent), while remaining completely free from additional optimization problems (e.g., from finding optimal projections, as in (Fayad and Ibrahim, 2023; Tsur et al., 2023)). Combined with fast convergence rates, this has established SMI as a scalable alternative to MI: computing the former typically requires orders of magnitude less time than neural MI estimation (several seconds vs. several hours for SOTA diffusion MI estimators (Franzese et al., 2024; Kholkin et al., 2025)). Consequently, it has been widely adopted for studying DNNs (Dentan et al., 2024; Wongso et al., 2022; 2023a; 2023b; 2025), deriving generalization bounds (Nadjahi et al., 2023), independence testing (Hu et al., 2024), auditing differential privacy (Nuradha and Goldfeld, 2023), feature selection (Goldfeld and Greenewald, 2021) and disentanglement in generative models (Goldfeld et al., 2022).

Despite its popularity, the research community has largely overlooked potential shortcomings of SMI. Some studies prematurely attribute their results to underlying phenomena without rigorously investigating whether they stem from artifacts introduced by random projections. Furthermore, existing works fail to comprehensively address issues related to random slicing, focusing primarily on suboptimality of random projections for information preservation (Fayad and Ibrahim, 2023; Tsur et al., 2023).

**Contribution.** In this article, we address this gap by systematically analyzing SMI across diverse settings, demonstrating that it frequently exhibits counterintuitive behavior and fails to accurately capture statistical dependence dynamics. Our key contributions are:

1. **Saturation and Sensitivity Analysis.** Our theoretical analysis and experiments reveal that SMI saturates prematurely, even for low-dimensional synthetic problems, and fails to detect significant increases in statistical dependence.

2. **Redundancy Bias.** We refute the prevailing assumption that SMI favors linearly extractable information by constructing an explicit example where introducing such structure increases MI and even linear correlation, but decreases SMI. In fact, we show that SMI prioritizes information *redundancy* over information content. We argue that this bias can lead to catastrophic failures in some applications.

3. **Curse of Dimensionality.** We revisit the dynamics of SMI for increasing dimensionality and argue that SMI is, in fact, cursed, with the curse of dimensionality manifesting itself not through sample complexity, but via asymptotic decay to zero in high-dimensional regimes due to diminishing redundancy.

4. **Reestablishing the Trade-off.** Finally, we discuss to which extent the aforementioned problems can be solved by using non-uniform/non-random slicing strategies, and how they affect the trade-off between scalability and utility.

In Section 2, we provide the necessary mathematical background. Section 3 overviews the related literature. Section 4 consists of our main theoretical results (see Section B for proofs). In Section 5, we employ synthetic benchmarks to show the disconnection between dynamics of MI and SMI. Sections 6 and 7 illustrate that SMI maximization may result in degenerate solutions, contrary to MI maximization. Finally, we discuss our results in Section 8.

## 2 Preliminaries

**Elements of Information Theory.** Let $(\Omega, \mathcal{F}, \mathbb{P})$ be a probability space with sample space $\Omega$, $\sigma$-algebra $\mathcal{F}$, and probability measure $\mathbb{P}$ defined on $\mathcal{F}$. Consider random vectors $X : \Omega \to \mathbb{R}^{d_x}$ and $Y : \Omega \to \mathbb{R}^{d_y}$ with joint distribution $\mathbb{P}_{X,Y}$ and marginals $\mathbb{P}_X$ and $\mathbb{P}_Y$, respectively. Wherever it is needed, we assume the relevant Radon-Nikodym derivatives exist. For any probability measure $\mathbb{Q}$ that is absolutely continuous w.r.t. $\mathbb{P}$ (denoted $\mathbb{Q} \ll \mathbb{P}$), the Kullback-Leibler (KL) divergence is $\mathsf{KL}[\mathbb{Q} \parallel \mathbb{P}] = \mathbb{E}_{\mathbb{Q}}\left[\log \frac{d\mathbb{Q}}{d\mathbb{P}}\right]$, which is non-negative and vanishes if and only if (iff) $\mathbb{P} = \mathbb{Q}$. The mutual information (MI) between $X$ and $Y$ quantifies the divergence between the joint distribution and the product of marginals:

$$\mathsf{I}(X;Y) = \mathbb{E}\log \frac{d\,\mathbb{P}_{X,Y}}{d\,\mathbb{P}_X \otimes \mathbb{P}_Y} = \mathsf{KL}\left[\mathbb{P}_{X,Y} \parallel \mathbb{P}_X \otimes \mathbb{P}_Y\right].$$

When $\mathbb{P}_X$ admits a probability density function (PDF) $p(X)$ with respect to (w.r.t.) the Lebesgue measure, the differential entropy is defined as $\mathsf{h}(X) = -\mathbb{E}[\log p(X)]$, where $\log(\cdot)$ denotes the natural logarithm. Likewise, the joint entropy $\mathsf{h}(X,Y)$ is defined via the joint density $p(X,Y)$, and conditional entropy is $\mathsf{h}(X \mid Y) = -\mathbb{E}[\log p(X \mid Y)] = -\mathbb{E}_Y\left[\mathbb{E}_{X\mid Y} \log p(X \mid Y)\right]$. Under the existence of PDFs, MI satisfies the identities

$$\mathsf{I}(X;Y) = \mathsf{h}(X) - \mathsf{h}(X \mid Y) = \mathsf{h}(Y) - \mathsf{h}(Y \mid X) = \mathsf{h}(X) + \mathsf{h}(Y) - \mathsf{h}(X,Y). \tag{2}$$

**Sliced Mutual Information.** In this work, we denote by $\mu_{\mathrm{M}}$ the normalized Haar (uniform) probability measure on a compact manifold $\mathrm{M}$, i.e., the unique bi-invariant measure satisfying $\mu_{\mathrm{M}}(\mathrm{M}) = 1$. Hence, to sample uniformly from specific spaces we write $\mathrm{W} \sim \mu_{\mathrm{O}(d)}, \theta \sim \mu_{\mathbb{S}^{d-1}}, \mathrm{A} \sim \mu_{\mathrm{St}(k,d)}$, indicating draws from the Haar measures on orthogonal group $\mathrm{O}(d) = \{\mathrm{Q} \in \mathbb{R}^{d \times d} : \mathrm{Q}^\mathsf{T}\mathrm{Q} = \mathrm{Q}\mathrm{Q}^\mathsf{T} = \mathrm{I}\}$, the unit sphere $\mathbb{S}^{d-1} = \{X \in \mathbb{R}^d : \|X\|_2 = 1\}$, and the Stiefel manifold $\mathrm{St}(k,d) = \{\mathrm{Q} \in \mathbb{R}^{d \times k} : \mathrm{Q}^\mathsf{T}\mathrm{Q} = \mathrm{I}\}$, respectively.

The $k$-sliced mutual information ($k$-SMI) (Goldfeld et al., 2022) between $X, Y$ is defined as

$$\mathsf{SI}_k(X;Y) = \int_{\mathrm{St}(k,d_x)} \int_{\mathrm{St}(k,d_y)} \mathsf{I}(\Theta^\mathsf{T}X; \Phi^\mathsf{T}Y)\, d\mu_{\mathrm{St}(k,d_x)}(\Theta)\, d\mu_{\mathrm{St}(k,d_y)}(\Phi),$$

Setting $k = 1$ recovers the standard sliced mutual information (1).

## 3 Background

Merits of SMI are straightforward and have been investigated thoroughly (Goldfeld et al., 2022; Goldfeld and Greenewald, 2021). We remind the reader of the most important of them:

1. **Scalability** enabled by low-dimensional projections.
2. **Nullification Property** (i.e., $\mathsf{SI}_k(X;Y) = 0$ iff $X$ and $Y$ are independent), which stems from the projections being random and independent.

In contrast, demerits of SMI are not very obvious and not well-covered in the literature. In this section, we recapitulate and analyze previous works which address the shortcomings of SMI. To facilitate the analysis, we divide them into three main categories.

**Suboptimality of random slicing.** Tsur et al. (2023) and Fayad and Ibrahim (2023) argue that a uniform slicing strategy can produce suboptimal projections, impairing SMI's ability to capture dependencies in the presence of noisy or non-informative components. To address this issue, Tsur et al. (2023) proposed max-sliced MI (mSMI), which selects non-random projectors that maximize the MI between projected representations. This approach is also claimed to improve interpretability and convergence rates.

However, deterministic slicing may overlook dependencies captured by non-optimal components. To mitigate this, Fayad and Ibrahim (2023) extend the max-sliced approach by optimizing SMI over probability distributions of projectors, with regularization to maintain slice diversity. While the authors emphasize that optimization should occur over *joint*

distributions, their motivation primarily addresses the issue of non-optimal *marginal* distributions of $\theta$ and $\phi$ — specifically, the presence of non-informative components in $X$ and $Y$. We contend that this represents only a partial understanding of the problem, as many SMI artifacts arise from other factors. Needless to say that optimization over probability distributions is also a heavy burden, which does not align with the slicing philosophy.

**Data Processing Inequality violation.** A fundamental property of MI is that it cannot be increased by deterministic or stochastic processing. Furthermore, MI is preserved under invertible transformations. This is formalized by the *data processing inequality* (DPI).

**Theorem 3.1.** (Polyanskiy and Wu (2024, Theorem 3.7)) For a Markov chain $X \to Y \to Z$, $\mathsf{I}(X;Y) \geq \mathsf{I}(X;Z)$. Additionally, if $Z = f(Y)$ where $f$ is invertible, then equality holds.

In contrast to MI, SMI violates the DPI (Goldfeld and Greenewald, 2021, Section 3.2). While the intuition behind DPI is clear (raw data already contains full information, and processing can only destroy it), the implications of DPI violation are less straightforward.

Existing works suggest that SMI's violation of DPI can reflect a preference for linearly extractable features, framing this as a useful property that aligns with the informal understanding of "practically available" (i.e., easily accessible) information (Goldfeld and Greenewald, 2021; Wongso et al., 2022; 2025). However, this interpretation can be misleading if the factors behind SMI increases are misidentified. Our analysis reveals that this is indeed the case, as SMI exhibits more inherent biases than previously recognized.

**Asymptotics in high-dimensional regime.** Convergence analysis suggests that the sample complexity of SMI estimation is far less sensitive to data dimensionality compared to that of MI. In fact, it has been argued that the estimation error may even decrease with dimensionality in some cases (Goldfeld et al., 2022, Remark 4). However, this behavior may result from SMI vanishing as dimensionality grows. Specifically, (Goldfeld et al., 2022, Theorem 3) provides an asymptotic expression (as $d \to \infty$) for SMI between jointly normal $X$ and $Y$, which decays hyperbolically with $d$ under some circumstances.

To date, no explanation for this phenomenon has been provided in the literature. We therefore elaborate on this finding by deriving non-asymptotic expressions, along with experimental results for non-Gaussian data, which reveal further nuances behind the decay.

## 4 THEORETICAL ANALYSIS

We start our analysis with considering a simple example, which (a) admits closed-form expression for SMI and (b) highlights severe problems of the quantity in question.

**Lemma 4.1.** Consider the following pair of jointly Gaussian $d$-dimensional random vectors:

$$(X, Y) \sim \mathcal{N}\left(0, \begin{pmatrix} \mathrm{I} & \rho\mathrm{I} \\ \rho\mathrm{I} & \mathrm{I} \end{pmatrix}\right), \quad \rho \in (-1; 1).$$

In this setup, MI and SMI can be calculated analytically:

$$\mathsf{I}(X;Y) = -\frac{d}{2}\log(1-\rho^2), \qquad \mathsf{SI}(X;Y) = \frac{\rho^2}{2d}\,_3F_2\left(1, 1, \frac{3}{2}; \frac{d}{2}+1, 2; \rho^2\right),$$

where $_3F_2$ is the *generalized hypergeometric function*. Additionally, the following limits hold:

$$\lim_{d\to\infty} \mathsf{I}(X;Y) = +\infty \qquad \lim_{d\to\infty} \mathsf{SI}(X;Y) = 0$$

$$\lim_{\rho^2\to 1} \mathsf{I}(X;Y) = +\infty \qquad \lim_{\rho^2\to 1} \mathsf{SI}(X;Y) = \psi(d-1) - \psi\left(\frac{d-1}{2}\right) - \log 2 \leq \frac{1}{d-1},$$

with $\psi$ being the *digamma function*.

Note that while MI correctly captures the growing statistical dependence as $d \to \infty$ (since additional components contribute shared information), SMI drops to zero, exposing a fundamental problem. We interpret this behavior as a distinct manifestation of the **curse**

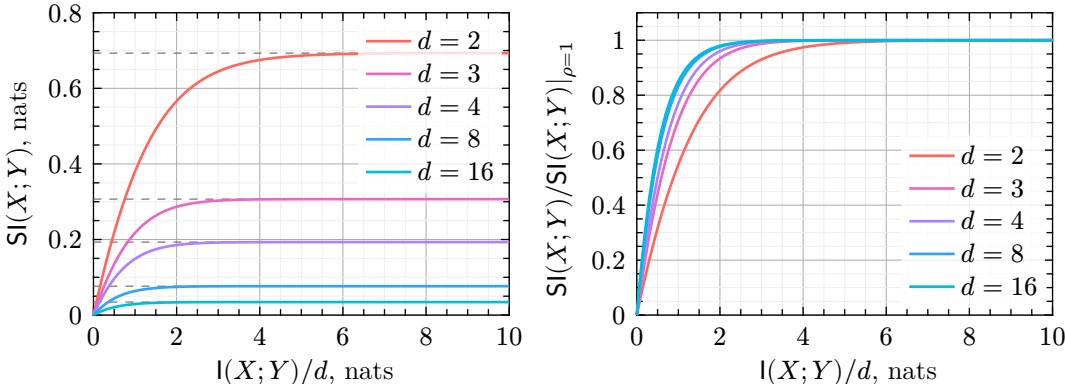

Figure 2: Saturation of $\mathsf{SI}(X;Y)$ as function of $\mathsf{I}(X;Y)/d$ for the example from Lemma 4.1, non-normalized (left) and normalized (right) versions. Note that the problem becomes more prominent in higher dimensions, both because of lower plateau and faster saturation.

**of dimensionality**: as $d$ grows, SMI uniformly decays to zero and becomes ineffective for statistical analysis. We also provide a less tight, but more general result in Section C.

The second pair of limits reveals another critical flaw of SMI. When $\rho^2 \to 1$, the $X$-$Y$ relationship becomes deterministic — a property MI reflects successfully. In stark contrast, SMI remains bounded by a dimension-dependent factor that decays hyperbolically. Furthermore, plotting SMI against MI shows this bound is reached prematurely, demonstrating SMI's **rapid saturation** with increasing dependence (Figure 2). In this saturated regime, SMI becomes effectively insensitive to further growth in shared information. Moreover, this renders estimates of SMI for different dimensionalities fundamentally incomparable, as they are theoretically bounded by factors depending on $d$.

These phenomena can not be explained by suboptimality of individual projections. In fact, each individual projection is optimal, as $\mathsf{I}(\theta^\mathsf{T} X;Y)$ does not depend on $\theta$ in this particular example. The proof of Lemma 4.1 suggests that the problem arises from the majority of *pairs* of projectors being suboptimal, yielding near-independent $\theta^\mathsf{T} X$ and $\phi^\mathsf{T} Y$ in the most outcomes, even for $d = 2$. Although similar analysis for $k$-SMI is extremely challenging, we argue that the problems in question prevail even when employing $k$-rank projectors.

**Proposition 4.2.** Under the setup of Lemma 4.1, $k$-SMI has the following representation

$$\mathsf{SI}_k(X;Y) = -\frac{1}{2}\int_{[0,1]^k} \sum_{i=1}^{k} \log(1-\rho^2\lambda_i)\, p(\boldsymbol{\lambda})\, \mathrm{d}\boldsymbol{\lambda}, \quad p(\boldsymbol{\lambda}) \propto \prod_{i<j}|\lambda_j - \lambda_i| \underbrace{\prod_{i=1}^{k}(1-\lambda_i)^{(d-2k-1)/2}}_{(\star)}$$

*Remark* **4.3.** As $d$ grows, $(\star)$ asymptotically concentrates $\lambda_i$ near zero, driving $\mathsf{SI}_k$ to zero.

We argue that the limitations we uncovered can be attributed to a strong bias of SMI toward **information redundancy**. That is, SMI favors repetition of information across different axes, and suffers from the curse of dimensionality if $X$ and $Y$ have high entropy. The following proposition and remark present a simple example to clarify this bias.

**Proposition 4.4.** Let $X$ and $Y$ be $d_x, d_y$-dimensional random vectors respectively, with $d_x, d_y < k$. Let $\mathrm{A} \in \mathbb{R}^{m_x \times d_x}$, $\mathrm{B} \in \mathbb{R}^{m_y \times d_y}$ be full column rank. Then $\mathsf{SI}_k(\mathrm{A}X;\mathrm{B}Y) = \mathsf{I}(X;Y)$.

**Corollary 4.5.** Consider the following pair of Gaussian $d$-dimensional random vectors:

$$(X, Y) \sim \mathcal{N}\left(0, \begin{pmatrix} \mathrm{J} & \rho\mathrm{J} \\ \rho\mathrm{J} & \mathrm{J} \end{pmatrix}\right), \quad \rho \in (-1;1),$$

where $\mathrm{J} = \mathbf{1} \cdot \mathbf{1}^\mathsf{T}$ with $\mathbf{1}^\mathsf{T} = (1, ..., 1)$. Then $\mathsf{SI}_k(X;Y) = \mathsf{I}(X;Y) = -\frac{1}{2}\log(1-\rho^2)$.

*Remark* **4.6.** Applying $\mathbf{1} \cdot e_1^\mathsf{T}$ to $X$ and $Y$ from Lemma 4.1 individually yields the example from Corollary 4.5. Therefore, this linear transform increases SMI despite decreasing MI.

### 4.1 Extension to optimal slicing

Although our work primarily focuses on conventional (average) SMI, as it is the most widely used variant, we also provide some intuition regarding the limitations of its "optimal" counterparts: *max-sliced* MI (mSMI) (Tsur et al., 2023) and *optimal-sliced* MI (oSMI) (Fayad and Ibrahim, 2023). Since mSMI is a special case of oSMI without regularization, we restrict our discussion to it, though our reasoning extends to oSMI as well. The $k$-mSMI is defined as:

$$\overline{\mathsf{SI}}_k(X;Y) = \sup_{\Theta \in \text{St}(d_x,k),\ \Phi \in \text{St}(d_y,k)} \mathsf{I}(\Theta^\mathsf{T} X; \Phi^\mathsf{T} Y). \tag{3}$$

The following proposition highlights the shortcomings of linear compression: even in a simple Gaussian setting, mSMI captures only a subset of dependencies and can exhibit opposite trends to MI. This occurs, for instance, when dependencies become more evenly distributed across components, which again returns us to the **redundancy bias**.

**Proposition 4.7.** (Tsur et al. (2023, Proposition 2)) Let $(X, Y) \sim \mathcal{N}(\mu, \Sigma)$, with marginal covariances $\Sigma_X$, $\Sigma_Y$ and cross-covariance $\Sigma_{XY}$. Suppose the matrix $\Sigma_X^{-\frac{1}{2}} \Sigma_{XY} \Sigma_Y^{-\frac{1}{2}}$ exists, and let $\{\rho_i\}_{i=1}^d$ denote its singular values in descending order, where $d = \min(d_x, d_y)$. Then

$$\mathsf{I}(X;Y) = -\frac{1}{2}\sum_{i=1}^d \log(1 - \rho_i^2), \qquad \overline{\mathsf{SI}}_k(X;Y) = -\frac{1}{2}\sum_{i=1}^k \log(1 - \rho_i^2).$$

## 5 Synthetic experiments

To complement our theoretical analysis and address complex, non-Gaussian cases, we conduct an extensive benchmarking of SMI using synthetic tests from (Butakov et al., n.d.), based on the works of (Butakov et al., 2024; Czyż et al., 2023). These tests are designed to evaluate MI estimators. However, here we do not assess whether SMI estimates converge to ground-truth MI values. SMI is a *distinct measure of statistical dependence*, and should not be viewed as an approximation of MI. Instead, our analysis focuses on the relationship between the two measures: since MI captures the true degree of statistical dependence, opposing trends in MI and SMI reveal problems with the latter quantity.

For the experiments, we use *correlated normal*, *correlated uniform*, *smoothed uniform* and *log-gamma-exponential* distributions, for which the ground-truth value of MI is available. To increase the dimensionality, we use independent components with equally distributed per-component MI. For each distribution, we vary both the data dimensionality ($d$) and the projection dimensionality ($k < d$). In Section G, we also utilize MI-preserving mappings to transform low-dimensional Gaussian vectors into high-dimensional synthetic images, as described in (Butakov et al., 2024); the examples of such images are displayed in Figure 3.

To estimate MI between projections, we use the KSG estimator (Kraskov et al., 2004) with the number of neighbors fixed at 1 (higher values are suboptimal, see Section F), $10^4$ samples from $(X, Y)$ and 128 samples from $(\Theta, \Phi)$. For each configuration, 10 independent runs with different random seeds are conducted to compute means and standard deviations.

To experimentally verify the saturation, we plot SMI against MI normalized by dimensionality $d$ in Figure 4. The plots clearly show that SMI reaches a plateau relatively early for all the featured distributions. The results for the normal distribution also align well with

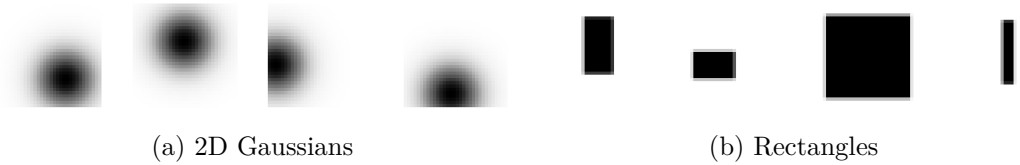

|              (a) 2D Gaussians              |              (b) Rectangles              |

Figure 3: Examples of synthetic images from additional experiments in Section G. Note that images are high-dimensional, but admit latent structure, which is similar to real datasets.

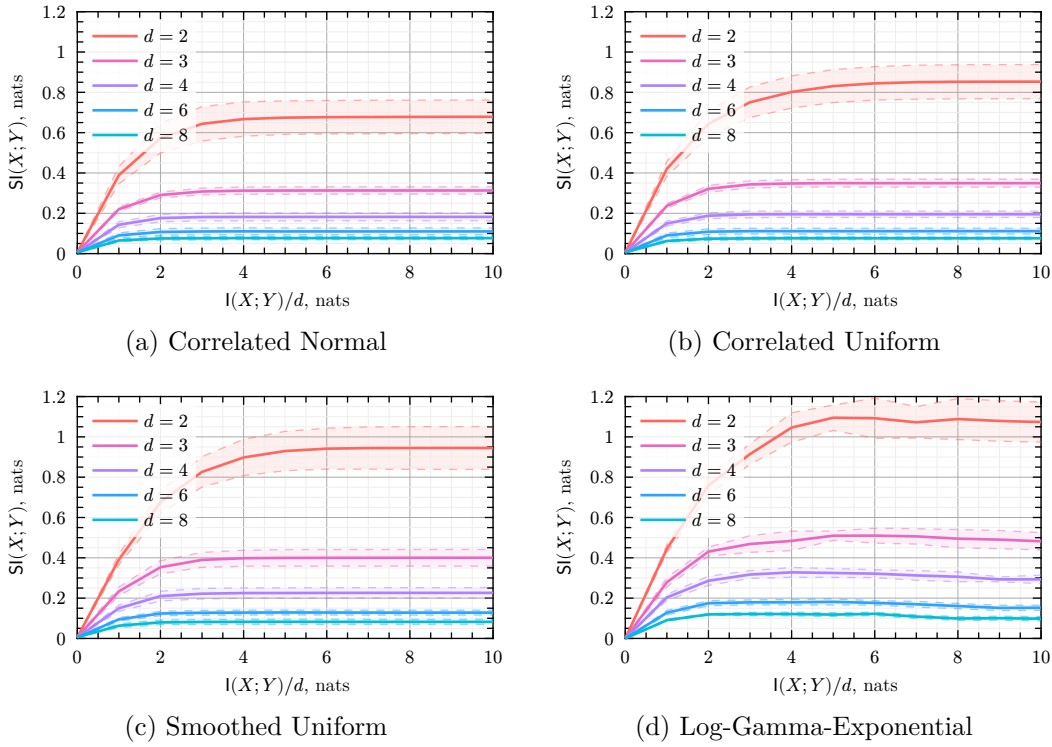

Figure 4: SMI results on synthetic benchmarks. Mean values and standard deviations across 10 runs are reported, $10^4$ samples from $X, Y$ and 128 random projections were used.

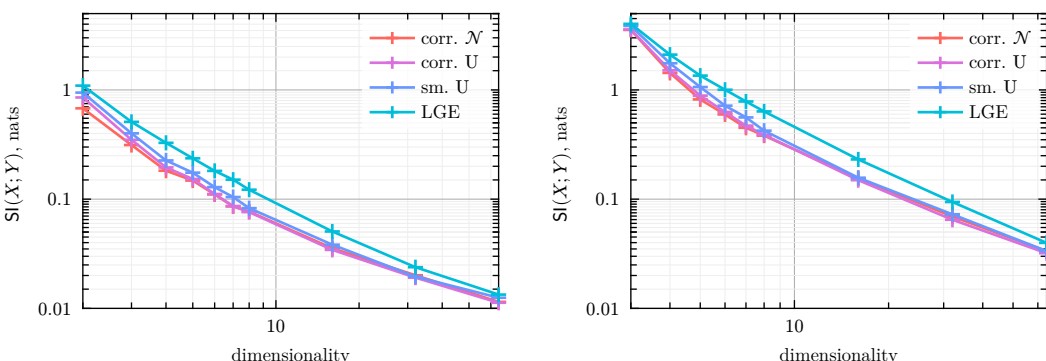

Figure 5: Saturated values of $k$-SMI versus data dimensionality $d$ for **1-SMI (left)** and **2-SMI (right)** for *correlated normal* (corr. $\mathcal{N}$), *correlated uniform* (corr. U), *smoothed uniform* (sm. U) and *log-gamma-exponential* (LGE).5 Log scale illustrates the $1/d$ trend.

those from Lemma 4.1. We further confirm the saturation of $k$-SMI for $k \in \{2,3\}$ and for complex datasets from (Butakov et al., 2024) experimentally in Section G. Finally, we plot the saturated values against $d$ on a log-log scale, demonstrating that the $1/d$ trend from Lemma 4.1 also holds for non-Gaussian distributions.

Overall, the results strongly support our findings, showing saturation and uniform decay with increasing dimensionality across a wide range of settings, from low-dimensional distributions to high-dimensional images.

## 6 SMI FOR INFOMAX-LIKE TASKS

Since mutual information is interpretable and captures non-linear dependencies, it is widely used as a training objective. Many applications involve maximizing MI (InfoMax) for feature

selection (Battiti, 1994; Sulaiman and Labadin, 2015; Vergara and Estévez, 2014; Yang and Gu, 2004) and self-supervised representation learning (Bachman et al., 2019; Butakov et al., 2025; Hjelm et al., 2019; Tschannen et al., 2020; Veličković et al., 2019). However, due to the curse of dimensionality, it was instead proposed to maximize SMI for feature extraction (Goldfeld and Greenewald, 2021) and disentanglement in InfoGAN (Goldfeld et al., 2022).

In this section, we argue that SMI is not a suitable alternative to MI for InfoMax tasks: since SMI exhibits a strong preference for redundancy, SMI maximization may lead to collapses.

**Representation learning.**   To demonstrate SMI's redundancy bias, we examine learning compressed representations through information maximization (*Deep InfoMax*) (Hjelm et al., 2019). This approach is known to be equivalent to many popular contrastive self-supervised methods (Butakov et al., 2025).

In Deep InfoMax, an encoder network $f$ is trained to maximize a lower bound on $\mathsf{I}(X; f(X))$, where $X$ represents input data and $f(X)$ its compressed representation. This method is theoretically sound, as maximizing MI ensures the most informative embeddings under the latent space dimensionality constraint. For our study, we replace MI with SMI in this framework. This substitution is straightforward since both MI and SMI admit Donsker-Varadhan variational lower bounds (Donsker and Varadhan, 1983):

$$\mathsf{I}(X;Y) = \sup_{T:\Omega\to\mathbb{R}} \Big[\mathbb{E}_{\mathbb{P}_{X,Y}} T(X,Y) - \log\big(\mathbb{E}_{\mathbb{P}_X \otimes \mathbb{P}_Y} e^{T(X,Y)}\big)\Big],$$
$$\mathsf{SI}_k(X;Y) = \sup_{T:\Omega\to\mathbb{R}} \mathbb{E}_{\Theta,\Phi}\Big[\mathbb{E}_{\mathbb{P}_{X,Y}} T(\Theta^\mathsf{T} X, \Phi^\mathsf{T} Y, \Theta, \Phi) - \log\big(\mathbb{E}_{\mathbb{P}_X \otimes \mathbb{P}_Y} e^{T(\Theta^\mathsf{T} X, \Phi^\mathsf{T} Y, \Theta, \Phi)}\big)\Big], \quad (4)$$

where $T$ is a critic function, which is also approximated in practice by a neural network. For detailed derivations of these bounds, we refer the reader to (Belghazi et al., 2018) (MI) and (Goldfeld et al., 2022; Goldfeld and Greenewald, 2021) (SMI).

We strictly follow the experimental protocol from (Butakov et al., 2025). In particular, we use MNIST handwritten digits dataset (Deng, 2012), employ InfoNCE loss (Oord et al., 2019) to approximate (4), use convolutional network for $f$ and fully-connected network for $T$. Latent space dimensionality is fixed at $d = 2$ for visualization purposes. Small Gaussian noise is added to the outlet of the encoder to combat representation collapse (Butakov et al., 2025). For more details, see Section I. We focus on this simple setup because our objective is to show that SMI produces degenerate results even in elementary tasks, making more complex configurations unnecessary for this demonstration.

Results are presented in Figure 6. As expected, maximization of SMI immediately leads to collapsed representations, while conventional InfoMax yields embeddings with low redundancy (their distribution is close to $\mathcal{N}(0, \mathbf{I})$). This behavior is consistent across different runs.

**Gaussian channel.**   We also refute SMI's preference for linearly extractable information by considering $X, Y$ such that $\operatorname{cov} X = \mathbf{I}$, $Y = \mathbf{A}X + \mathcal{N}(0, \sigma^2 \mathbf{I})$, and $\operatorname{diag} \mathbf{A}\mathbf{A}^\mathsf{T} = \mathbf{I}$; this is a Gaussian channel with energy constraints (Cover and Thomas, 2006). Generally, $\mathsf{I}(X; Y)$ is

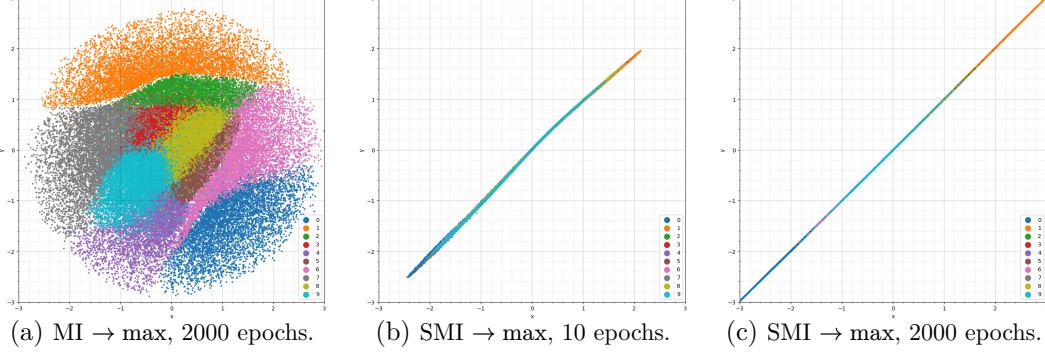

(a) MI → max, 2000 epochs.    (b) SMI → max, 10 epochs.    (c) SMI → max, 2000 epochs.

Figure 6: Visualizations of embeddings from the representation learning experiments, with points colored by class. Note that mutual information maximization (left) produces clustered low-redundancy representations, while SMI maximization results in immediate collapse.

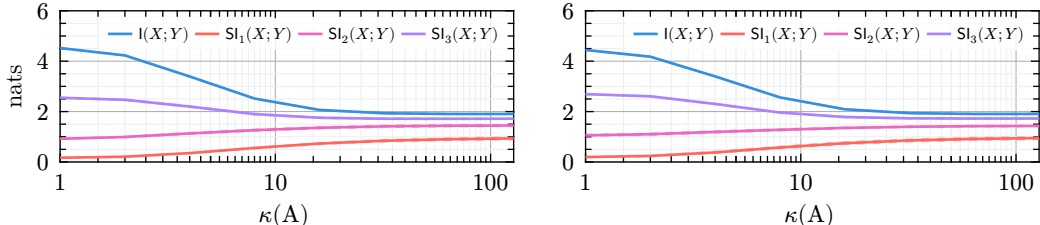

Figure 7: Changing the condition number of A in the Gaussian channel experiment ($Y = \mathbf{A}X + \mathcal{N}(0, \sigma^2 \mathbf{I}_d)$) for **normal** $X \sim \mathcal{N}(0, \mathbf{I}_d)$ **(left)** and **uniform** $X \sim \mathrm{U}[0; \sqrt{12}]^d$ **(right)**. We perform 10 runs with $10^4$ samples, 128 projections, and use $\sigma = 0.3$, $d = 4$.

maximized by a well-posed A, since decorrelated features are more robust to isotropic noise. However, the results in Figure 7 highlight SMI's preference for ill-posed A (i.e., matrices with high condition number $\kappa(\mathbf{A}) \overset{\text{def}}{=} \|\mathbf{A}\| \cdot \|\mathbf{A}^{-1}\|$). More information is in Section D.

# 7 REPLICATION STUDY

Since our work highlights fundamental problems with SMI, we revisit the experiments from the original SMI articles (Goldfeld et al., 2022; Goldfeld and Greenewald, 2021; Tsur et al., 2023) to reassess their results. We are especially interested in the **feature extraction** and **independence testing**, because these setups might suffer from the redundancy bias and SMI's decay to zero. Section H provides more details.

**Feature extraction.** In (Goldfeld and Greenewald, 2021), the following toy problem is considered: $\mathsf{SI}(\mathbf{A}X; \mathbf{B}Y) \to \max_{\mathbf{A},\mathbf{B}}$, where $X \sim \mathcal{N}(0, \mathbf{I}_d)$, $Y = \mathbf{1} \cdot e_1^\top X + \mathcal{N}(0, \mathbf{I}_d)$, and $\mathbf{A}, \mathbf{B} \in \mathbb{R}^{d \times d}$ are feature selection matrices. The redundancy bias suggests that optimal $\mathbf{A}, \mathbf{B}$ are singular, with all columns other than the first being zero — a property reflected in the original results (Goldfeld and Greenewald, 2021, Figure 3).

To highlight that SMI fails when the number of relevant features increases, we consider the following example: $X \sim \mathcal{N}(0, \mathbf{I}_d)$, $Y = \sum_{i=1}^m e_i X_i + \mathcal{N}(0, \mathbf{I}_d)$, where $m$ controls the number of features. In addition to maximizing $k$-SMI, we also learn $\mathbf{A}, \mathbf{B}$ through MI maximization. In the latter case, we use $\mathbf{A}, \mathbf{B} \in \mathbb{R}^{k \times d}$ to impose a dimensionality bottleneck. For MI and $k$-SMI maximization, we reuse the NNs from Section 6 and perform 10 runs with $10^4$ samples.

The quality of feature extraction is assessed via the *effective rank* (Roy and Vetterli, 2007) of the matrices formed by the first $m$ columns of A and B respectively. Figure 9 illustrates that MI maximization yields effective rank close to $k$, confirming its ability to recover all relevant features. In contrast, $k$-SMI results in a low effective rank regardless of $k$, revealing its redundancy bias. A visual analysis of the matrices in Figure 10 and Section H.1 also supports our findings.

**Independence testing.** Goldfeld et al. (2022); Goldfeld and Greenewald (2021) report consistently superior performance of SMI over MI for independence testing when the data dimensionality $d$ is fixed. We replicate their protocol for the distributions from Section 5 but introduce a critical modification. Instead of evaluating each $d$ separately, we pool SMI (and MI) estimates across multiple dimensions ($d \in \{2, 10, 20, 30\}$) for each sample size $n$ and compute a single ROC-AUC from the mixed-dimensional data. For a fair comparison,

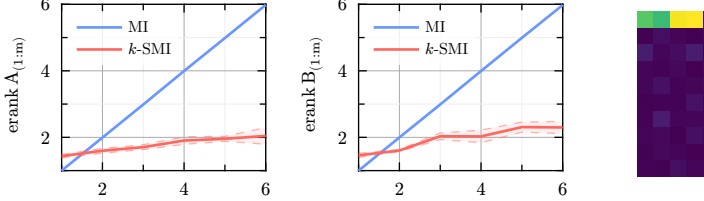

Figure 9: Effective rank versus $k$ for feature extraction; 10 runs with $10^4$ samples, $m = 6$.

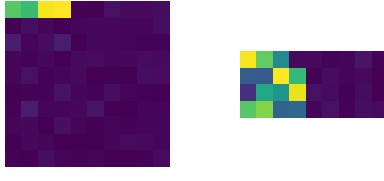

Figure 10: Matrices A for SMI $\to$ max (left) and MI $\to$ max (right), $m = k = 4$.

$I(X; Y)$ is fixed at $2$ nat, and KSG (Kraskov et al., 2004) is used as a backbone MI estimator. We conduct $100$ runs for each $d$.

As shown in Figure 11, and in contrast to (Goldfeld et al., 2022; Goldfeld and Greenewald, 2021), SMI performs worse under this more realistic setting where a single threshold must work across varying dimensions. These experiments reveal that SMI's discriminative power can drop sharply even when the ground truth MI is constant, causing dependent high-dimensional cases to yield SMI values that overlap with independent low-dimensional cases. Consequently, it is hard to consider SMI reliable enough for independence testing, unless the dimensionality is fixed in advance.

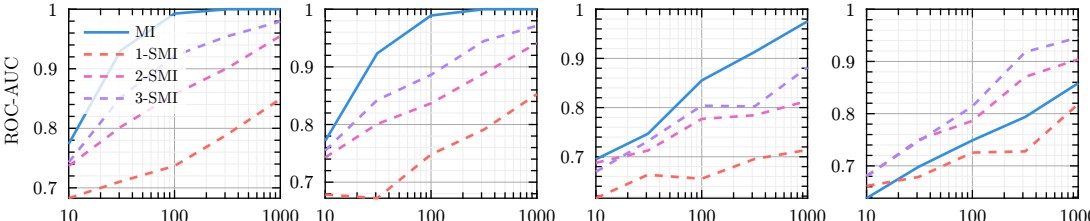

Figure 11: Independence testing: ROC-AUC versus sample size for correlated normal, correlated uniform, smoothed uniform and log-gammma-exponential (left-to-right, $2$ nat).

## 8 Discussion

**Results.** Sliced mutual information (SMI) has been proposed as a scalable alternative to Shannon's mutual information. While SMI enables efficient computation in high-dimensional settings and satisfies the nullification property, our findings reveal critical deficiencies that undermine its reliability for feature extraction and related tasks.

We demonstrate that SMI saturates rapidly, failing to capture variations in statistical dependence. This makes it difficult to distinguish between intrinsic SMI fluctuations and genuine changes in dependence structure. Furthermore, we invalidate the common hypothesis that SMI favors linear features through a counterexample where even correlation coefficients reflect dependence more faithfully than SMI, which exhibits inverted behavior.

In high dimensions, SMI decays with increasing dimensionality, contrary to MI's monotonic behavior. This is established analytically for Gaussian cases and validated empirically across diverse synthetic experiments. Consequently, SMI variations may reflect redundancy or high-dimensional artifacts without a principled way to disentangle these factors.

**Impact.** Thanks to fast convergence rates and the absence of additional optimization problems, SMI has been widely applied across various fields of statistics and machine learning. Given our findings, it is therefore crucial to recognize how the inherent biases of SMI affect practical applications.

The works (Chen et al., 2023; Goldfeld et al., 2022; Goldfeld and Greenewald, 2021) propose using SMI in a Deep InfoMax setting. However, we demonstrate that maximizing SMI can lead to collapsed solutions due to the redundancy bias. Meanwhile, (Dentan et al., 2025; Shaeri and Middel, 2025; Wongso et al., 2022; 2023b; 2023a; 2025) study deep neural networks by measuring SMI between intermediate layers. Yet, as our analysis reveals, changes in SMI do not always reflect true shifts in statistical dependence; they may instead result from differences in layer dimensionality, redundancy in intermediate representations, low sensitivity in saturated regimes, or other factors. Finally, (Nuradha and Goldfeld, 2023) suggests using SMI for independence testing in differential privacy tasks. We contend that this approach poses critical issues, as SMI estimates can become statistically indistinguishable from zero in high-dimensional or low-redundancy settings.

**Limitations.** While we support our claims with both theoretical analysis and experimental evidence, we were able to derive precise analytical expressions for the Gaussian case only. Nevertheless, our findings are more than sufficient to expose fundamental limitations of SMI, and to support all the claims we made.

**Acknowledgments.** The work was supported by the grant for research centers in the field of AI provided by the Ministry of Economic Development of the Russian Federation in accordance with the agreement 000000C313925P4F0002 and the agreement №139-10-2025-033.

**Ethics statement.** This work is not subject to any ethical concerns.

**Reproducibility statement.** To ensure reproducibility of our results, we provide complete proofs in Section B and implementation details in Section I. We also provide our code for the experiments in the supplementary material.

**LLM usage.** Large Language Models (LLMs) were used only to assist with rephrasing sentences and improving the clarity of the text.

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

# A SUPPLEMENTARY THEORY

**Lemma A.1.** (Polyanskiy and Wu (2024, Example 2.4)) $h(\mathcal{N}(\mu, \Sigma)) = \frac{1}{2} \log\big((2\pi e)^d \det \Sigma\big).$

**Corollary A.2.** For $(X, Y) \sim \mathcal{N}(\mu, \Sigma)$ with non-singular $\Sigma$

$$I(X;Y) = \frac{1}{2} \log \det \Sigma_X + \frac{1}{2} \log \det \Sigma_Y - \frac{1}{2} \log \det \Sigma$$

$$= -\frac{1}{2} \sum_{i=1}^{d} \log(1 - \rho_i^2),$$

where $\Sigma_X$, $\Sigma_Y$ are marginal covariances, $\Sigma_{XY}$ is cross-covariance, $d = \min(d_x, d_y)$, and $\{\rho_i\}_{i=1}^{d}$ are singular values of $\Sigma_X^{-\frac{1}{2}} \Sigma_{XY} \Sigma_Y^{-\frac{1}{2}}$.

*Proof of Corollary A.2.* Combining Lemma A.1 and (2) yields the first result. Now note that

$$I(X;Y) = I\left(\Sigma_X^{-\frac{1}{2}} X; \Sigma_Y^{-\frac{1}{2}} Y\right) = I\left(U^\mathsf{T} \Sigma_X^{-\frac{1}{2}} X; V \Sigma_Y^{-\frac{1}{2}} Y\right),$$

where $U \operatorname{diag}(\rho_i) V^\mathsf{T}$ is the SVD of $\Sigma_X^{-\frac{1}{2}} \Sigma_{XY} \Sigma_Y^{-\frac{1}{2}}$. Now note that

$$\left(U^\mathsf{T} \Sigma_X^{-\frac{1}{2}} X, V \Sigma_Y^{-\frac{1}{2}} Y\right) \sim \mathcal{N}\left(\mu', \begin{pmatrix} I & \operatorname{diag}(\rho_i) \\ \operatorname{diag}(\rho_i) & I \end{pmatrix}\right),$$

from which we arrive at the second expression. $\qquad\square$

**Lemma A.3.** Let $A \in \mathbb{R}^{n \times m}$ be full column rank matrix, and $\Theta \sim \mu_{\mathrm{St}(n,k)}$, where $k > m$. Then $\Theta^\mathsf{T} A$ is full-rank with probability one.

*Proof of Lemma A.3.* Performing QR decomposition of $A$ yields $\Theta^\mathsf{T} A = \Theta^\mathsf{T} QR \overset{\mathrm{d}}{=} \Theta^\mathsf{T} \begin{pmatrix} I_m \\ 0 \end{pmatrix} R$. Since $A$ is full-rank, $R$ is invertible and $\operatorname{rank} \Theta^\mathsf{T} A = \operatorname{rank} \Theta^\mathsf{T} \begin{pmatrix} I_m \\ 0 \end{pmatrix}$. Therefore,

$$\mathbb{P}\{\Theta^\mathsf{T} A \text{ is full-rank}\} = 1 - \mathbb{P}\left\{\Theta^\mathsf{T} \begin{pmatrix} I_m \\ 0 \end{pmatrix} \text{ is not full-rank}\right\} = 1.$$

$\qquad\square$

**Lemma A.4.** (Edelman and Sutton (2008, Theorem 1.5)) Let $W \sim \mu_{\mathrm{O}(d)}$ and partition

$$W = \begin{pmatrix} W_{11} & W_{12} \\ W_{21} & W_{22} \end{pmatrix}$$

with $W_{11}$ of size $k$ by $k$. Then the eigenvalues $\{\lambda_i\}_{i=1}^{k}$ of $W_{11} W_{11}^\mathsf{T}$ follow the Jacobi ensemble

$$p(\boldsymbol{\lambda}) \propto \prod_{i<j} |\lambda_i - \lambda_j|^\beta \prod_{i=1}^{k} \lambda_i^{\frac{\beta}{2}(a+1)-1} (1 - \lambda_i)^{\frac{\beta}{2}(b+1)-1}$$

with parameters $a = 0, b = d - 2k$, and $\beta = 1$ (over $\mathbb{R}$).

*Proof of Lemma A.4.* Let $A_1 \in \mathbb{R}^{k \times d}$ and $A_2 \in \mathbb{R}^{(d-k) \times d}$ be independent matrices with i.i.d. entries from $\mathcal{N}(0, 1)$. By stacking $A_1$ atop $A_2$ and then performing a QR decomposition on the resulting Gaussian matrix, the orthogonal invariance of the Gaussian law implies that $Q$ is independent of the upper-triangular factor $R$ and uniformly distributed on $\mathrm{O}(d)$.

For a matrix $P = \operatorname{diag}(p_1, ..., p_k)$ with i.i.d. $p_i$ sampled uniformly from $\{-1, 1\}$, we have $QP \overset{\mathrm{d}}{=} W$. Partitioning $Q$ and $P$ into blocks similarly to $W$, we have $Q_{11} P_{11} \overset{\mathrm{d}}{=} W_{11}$ for the top-left block of $Q$.

The CS decomposition of an orthogonal $Q$ together with invertible $R$ yields the generalized singular value decomposition (GSVD) of the pair $(A_1, A_2)$:

$$\begin{pmatrix} A_1 \\ A_2 \end{pmatrix} = \begin{pmatrix} Q_{11} & Q_{12} \\ Q_{21} & Q_{22} \end{pmatrix} R = \begin{pmatrix} U_1 & \\ & U_2 \end{pmatrix} \begin{pmatrix} C & S & \\ -S & C & \\ & & I \end{pmatrix} \begin{pmatrix} V_1^\mathsf{T} & \\ & V_2^\mathsf{T} \end{pmatrix} R,$$

where $U_1, V_1 \in O(k), U_2, V_2 \in O(d-k)$, and $C = \mathrm{diag}(c_1,...,c_k)$, $S = \mathrm{diag}(s_1,...,s_k)$ with $c_i \geq 0$, $s_i \geq 0$ in descending order, and $c_i^2 + s_i^2 = 1$ for all $i$. The diagonal entries of $C$ are known as the generalized singular values of the pair $(A_1, A_2)$. From this decomposition and the SVD of $W_{11} = U\Sigma V^\mathsf{T}$, one has

$$U_1 C V_1^\mathsf{T} P_{11} \overset{d}{=} U\Sigma V^\mathsf{T}.$$

Since $U_1, V_1$, and $U, V$ are uniformly distributed on $O(k)$ and independent of $C, \Sigma, P_{11}$, we have $C \overset{d}{=} \Sigma$ by the invariance of the Haar measure under orthogonal transformations. On the other hand, the generalized singular values $\{c_i\}_{i=1}^k$ of a pair $(A_1, A_2)$ follow the law of the Jacobi ensemble with parameters $a = 0, b = d - 2k$, and $\beta = 1$ (Edelman and Sutton, 2008, Proposition 1.2). Therefore, the squared singular values of $W_{11}$ follow the Jacobi ensemble with the same parameters. $\square$

**Corollary A.5.** The squared inner product $|\theta^\mathsf{T}\phi|^2$ between two independent random vectors $\theta, \phi \sim \mu_{\mathbb{S}^{d-1}}$ follows $\mathrm{Beta}\left(\frac{1}{2}, \frac{d-1}{2}\right)$. Moreover, the shifted inner product $(1 + \theta^\mathsf{T}\phi)/2$ is symmetrically distributed as $\mathrm{Beta}\left(\frac{d-1}{2}, \frac{d-1}{2}\right)$.

*Proof of Corollary A.5.* Setting Jacobi parameters $k = 1, a = 0, b = d - 2$ and $\beta = 1$, the density is proportional to $x^{-1/2}(1-x)^{(d-3)/2}$ on $[0,1]$, which matches the $\mathrm{Beta}\left(\frac{1}{2}, \frac{d-1}{2}\right)$ distribution.

Next, observe that $\theta^\mathsf{T}\phi$ has a density proportional to $(1-t)^{\frac{d-3}{2}}$ for $t \in [-1,1]$. Under the change of variables $\eta \sim \mathrm{Beta}\left(\frac{d-1}{2}, \frac{d-1}{2}\right)$.

$\square$

## B    Complete proofs

**Lemma 4.1.** Consider the following pair of jointly Gaussian $d$-dimensional random vectors:

$$(X, Y) \sim \mathcal{N}\left(0, \begin{pmatrix} I & \rho I \\ \rho I & I \end{pmatrix}\right), \quad \rho \in (-1; 1).$$

In this setup, MI and SMI can be calculated analytically:

$$\mathsf{I}(X;Y) = -\frac{d}{2}\log(1 - \rho^2), \qquad \mathsf{SI}(X;Y) = \frac{\rho^2}{2d} \,_3F_2\left(1, 1, \frac{3}{2}; \frac{d}{2} + 1, 2; \rho^2\right),$$

where $_3F_2$ is the *generalized hypergeometric function*. Additionally, the following limits hold:

$$\lim_{d\to\infty} \mathsf{I}(X;Y) = +\infty \qquad \lim_{d\to\infty} \mathsf{SI}(X;Y) = 0$$

$$\lim_{\rho^2\to1} \mathsf{I}(X;Y) = +\infty \qquad \lim_{\rho^2\to1} \mathsf{SI}(X;Y) = \psi(d-1) - \psi\left(\frac{d-1}{2}\right) - \log 2 \leq \frac{1}{d-1},$$

with $\psi$ being the *digamma function*.

*Proof of Lemma 4.1.* One can acquire $\mathsf{I}(X;Y) = -\frac{d}{2}\log(1 - \rho^2)$ from a general expression for MI of two jointly Gaussian random vectors (see Corollary A.2).

Recall that $(\theta^\mathsf{T}X, \phi^\mathsf{T}Y)$ is also Gaussian with cross-covariance $\rho\,\theta^\mathsf{T}\phi$. Therefore, by Corollary A.2 we have

$$\mathsf{SI}(X;Y) = \mathsf{I}\left(\theta^\mathsf{T}X; \phi^\mathsf{T}Y \mid \theta, \varphi\right) = -\frac{1}{2}\mathbb{E}[\log(1 - \rho^2\,|\theta^\mathsf{T}\phi|^2)].$$

From Corollary A.5, we note that $|\theta^\mathsf{T}\phi|^2 \sim \mathrm{Beta}\left(\frac{1}{2}, \frac{d-1}{2}\right)$, so

$$\mathsf{SI}(X;Y) = -\frac{1}{2\mathrm{B}\left(\frac{1}{2},\frac{d-1}{2}\right)} \int_0^1 \log(1-\rho^2 x)(1-x)^{\frac{d-3}{2}} x^{-\frac{1}{2}} \, \mathrm{d}x$$

$$= \frac{\rho^2}{2} \frac{\Gamma\left(\frac{d}{2}\right)}{\Gamma\left(\frac{1}{2}\right)\Gamma\left(\frac{d-1}{2}\right)} \int_0^1 x^{\frac{1}{2}}(1-x)^{\frac{d-3}{2}} \, {}_2F_1(1,1;2;\rho^2 x) \, \mathrm{d}x,$$

where the last equality follows from the identity $\log(1-z) = -z \, {}_2F_1(1,1;2;z)$ with hypergeometric function ${}_2F_1$. Appling Euler's integral transform (McBride, 1999, Eq. (2.2.3)) gives

$$\mathsf{SI}(X;Y) = \frac{\rho^2}{2d} \frac{\Gamma\left(\frac{d}{2}+1\right)}{\Gamma\left(\frac{3}{2}\right)\Gamma\left(\frac{d-1}{2}\right)} \int_0^1 x^{\frac{3}{2}-1}(1-x)^{\left(\frac{d}{2}+1\right)-\frac{3}{2}-1} \, {}_2F_1(1,1;2;\rho^2 x) \, \mathrm{d}x$$

$$= \frac{\rho^2}{2d} \, {}_3F_2\left(1,1,\frac{3}{2};\frac{d}{2}+1,2;\rho^2\right).$$

Here ${}_3F_2$ denotes the generalized hypergeometric function.

Finally, we calculate the limit of $\mathsf{SI}(X;Y)$ as $\rho^2 \to 1$ using properties of beta-distribution. Denoting $\eta = (1+\theta^\mathsf{T}\phi)/2 \sim \mathrm{Beta}\left(\frac{d-1}{2},\frac{d-1}{2}\right)$ (see Corollary A.5), we get

$$\mathsf{SI}(X;Y) = -\log 2 - \mathbb{E}\log(1-\eta) = -\log 2 - \mathbb{E}\log\eta = \psi(d-1) - \psi\left(\frac{d-1}{2}\right) - \log 2,$$

where $\psi$ is the digamma function. Using the bounds on digamma function (Elezovic et al., 2000), we get

$$\log\left(x+\frac{1}{2}\right) - \frac{1}{x} \le \psi(x) \le \log\left(x+e^{\psi(1)}\right) - \frac{1}{x}, \tag{5}$$

we derive an upper bound on this expression:

$$\psi(d-1) - \psi\left(\frac{d-1}{2}\right) - \log 2 \le \frac{1}{d-1} + \log\left(1 + \frac{e^{\psi(1)}-1}{d}\right)$$

To simplify the bound, one can note that $e^{\psi(1)} - 1 < 0$, as $\psi(1) < 0$. $\qquad\square$

**Proposition 4.2.** Under the setup of Lemma 4.1, $k$-SMI has the following representation

$$\mathsf{SI}_k(X;Y) = -\frac{1}{2}\int_{[0,1]^k} \sum_{i=1}^k \log(1-\rho^2\lambda_i)\, p(\boldsymbol{\lambda})\, \mathrm{d}\boldsymbol{\lambda}, \quad p(\boldsymbol{\lambda}) \propto \prod_{i<j}|\lambda_j-\lambda_i| \underbrace{\prod_{i=1}^k (1-\lambda_i)^{(d-2k-1)/2}}_{(\star)}$$

*Proof of Proposition 4.2.* Let $\mathsf{Q}_\mathsf{X}, \mathsf{Q}_\mathsf{Y} \sim \mu_{\mathrm{St}(k,d)}$. Then $\left[\mathsf{Q}_\mathsf{X}^\mathsf{T}X, \mathsf{Q}_\mathsf{Y}^\mathsf{T}Y\right] \sim \mathcal{N}(0,\Sigma)$, where $\Sigma$ is a $2k \times 2k$ covariance matrix with the following block structure

$$\Sigma = \begin{pmatrix} \mathsf{I}_k & \rho\,\mathsf{Q}_\mathsf{X}^\mathsf{T}\mathsf{Q}_\mathsf{Y} \\ \rho\,\mathsf{Q}_\mathsf{Y}^\mathsf{T}\mathsf{Q}_\mathsf{X} & \mathsf{I}_k \end{pmatrix}.$$

Using the formula for the determinant of a block matrix $\Sigma$ yields

$$\mathsf{SI}_k(X;Y) = -\frac{1}{2}\mathbb{E}[\log\det(\Sigma)] = -\frac{1}{2}\mathbb{E}\left[\log\det\left(\mathsf{I} - \rho^2\left(\mathsf{Q}_\mathsf{X}^\mathsf{T}\mathsf{Q}_\mathsf{Y}\right)\left(\mathsf{Q}_\mathsf{X}^\mathsf{T}\mathsf{Q}_\mathsf{Y}\right)^\mathsf{T}\right)\right].$$

By the invariance of the Haar measure under left and right multiplication, $\mathsf{Q}_\mathsf{X}^\mathsf{T}\mathsf{Q}_\mathsf{Y} \overset{d}{=} \mathsf{W}_{11}$, where $\mathsf{W}_{11}$ is a $k$ by $k$ left upper block of the matrix $\mathsf{W} \sim \mu_{\mathrm{O}(d)}$. According to Lemma A.4, the eigenvalues of $\mathsf{W}_{11}\mathsf{W}_{11}^\mathsf{T}$ follow Jacobi ensemble with parameters $a=0, b=d-2k$ and $\beta=1$:

$$p(\lambda) \propto \prod_{i<j}|\lambda_j-\lambda_i| \prod_{i=1}^k (1-\lambda_i)^{\frac{d-2k-1}{2}}.$$

Thus, we get a general expression for $k$-SMI

$$\mathsf{SI}_k(X;Y) = -\frac{1}{2}\int_{[0,1]^k}\sum_{i=1}^{k}\log(1-\rho^2\lambda_i)p(\lambda)\,\mathrm{d}\lambda.$$

$\square$

**Proposition 4.4.** Let $X$ and $Y$ be $d_x, d_y$-dimensional random vectors respectively, with $d_x, d_y < k$. Let $\mathrm{A} \in \mathbb{R}^{m_x \times d_x}$, $\mathrm{B} \in \mathbb{R}^{m_y \times d_y}$ be full column rank. Then $\mathsf{SI}_k(\mathrm{A}X; \mathrm{B}Y) = \mathsf{I}(X;Y)$.

*Proof of Proposition 4.4.* Using Lemma A.3 and $d_x, d_y < k$, we get that $\Theta^\mathsf{T}\mathrm{A}$ and $\Phi^\mathsf{T}\mathrm{B}$ are injective with probability one for independent $\Theta, \Phi$ distributed uniformly on $\mathrm{St}(d_x, k)$ and $\mathrm{St}(d_y, k)$. Therefore, according to Theorem 3.1, $[\mathsf{I}(\Theta^\mathsf{T}\mathrm{A}X; \Phi^\mathsf{T}\mathrm{B}Y) \mid \Theta, \Phi] = \mathsf{I}(X;Y)$ almost sure. As a result, $\mathsf{SI}_k(\mathrm{A}X; \mathrm{B}Y) = \mathsf{I}(\Theta^\mathsf{T}\mathrm{A}X; \Phi^\mathsf{T}\mathrm{B}Y \mid \Theta, \Phi) = \mathsf{I}(X;Y)$. $\square$

**Proposition 4.7.** (Tsur et al. (2023, Proposition 2)) Let $(X, Y) \sim \mathcal{N}(\mu, \Sigma)$, with marginal covariances $\Sigma_X$, $\Sigma_Y$ and cross-covariance $\Sigma_{XY}$. Suppose the matrix $\Sigma_X^{-\frac{1}{2}}\Sigma_{XY}\Sigma_Y^{-\frac{1}{2}}$ exists, and let $\{\rho_i\}_{i=1}^{d}$ denote its singular values in descending order, where $d = \min(d_x, d_y)$. Then

$$\mathsf{I}(X;Y) = -\frac{1}{2}\sum_{i=1}^{d}\log(1-\rho_i^2), \qquad \overline{\mathsf{SI}}_k(X;Y) = -\frac{1}{2}\sum_{i=1}^{k}\log(1-\rho_i^2).$$

*Proof of Proposition 4.7.* Direct corollary of Corollary A.2. $\square$

## C  GENERAL CASE

While Lemma 4.1 successfully demonstrates severe shortcomings in SMI, it relies exclusively on the Gaussian case. Since real-world data distributions can deviate significantly from normality, this section analyzes other scenarios where SMI may or may not exhibit limitations.

We begin with a simple example of discrete random vectors $X, Y$ for which $\mathsf{SI}_k(X;Y) = \mathsf{I}(X;Y)$ regardless of $k$ and dimensionality.

***Example* C.1.** Let $X, Y$ be any discrete pair random vectors. Then, $\mathsf{SI}_k(X;Y) = \mathsf{I}(X;Y)$.

*Proof of Example C.1.* Because $X$ and $Y$ are discrete, almost every random projection mapping is injective on their respective supports. Since MI is invariant under measurable injective transforms, $\mathsf{I}(\Theta^\mathsf{T}X; \Phi^\mathsf{T}Y) = \mathsf{I}(X;Y)$ for almost all fixed $\Theta$ and $\Phi$. Therefore, taking the expectations over $\Phi \sim \mu_{\mathrm{St}(k, d_X)}$, $\Theta \sim \mu_{\mathrm{St}(k, d_Y)}$ yields

$$\mathsf{SI}_k(X;Y) = \mathsf{I}(\Theta^\mathsf{T}X; \Phi^\mathsf{T}Y \mid \Theta, \Phi) = \mathsf{I}(X;Y).$$

$\square$

However, this example is simple and does not require dimensionality reduction in the first place: when dealing with discrete random vectors, the only constraint is the support size. On the other hand, applying SMI to continuous distributions with independent components immediately results in saturation.

**Lemma C.2.** Let $X : \Omega \to \mathbb{R}^d$ be a random vector with i.i.d. components of unit variance such that $\mathsf{h}(X_i) = E < \infty$, and $Y \perp\!\!\!\perp X_i$ for $i \geq 2$. Then

$$\mathsf{SI}_k(X;Y) \leq k(\mathsf{h}(\mathcal{N}(0,1)) - E) - \frac{1}{2}\left[\psi\left(\frac{d-k}{2}\right) - \psi\left(\frac{d}{2}\right)\right],$$

where the RHS is independent of $\mathsf{I}(X;Y)$.

*Proof of Lemma C.2.* From the DPI for the Markov chain $\Phi^\mathsf{T}Y \to Y \to X_1 \to \Theta^\mathsf{T}X$, one has

$$\mathsf{I}(\Theta^\mathsf{T} X; \Phi^\mathsf{T} Y \mid \Theta, \Phi) \leq \mathsf{I}(\Theta^\mathsf{T} X; X_1 \mid \Theta)$$
$$= \mathsf{h}(\Theta^\mathsf{T} X \mid \Theta) + \mathsf{h}(X_1) - \mathsf{h}(\Theta^\mathsf{T} X, X_1 \mid \Theta).$$

The first one can be upper bounded as follows

$$\mathsf{h}(\Theta^\mathsf{T} X \mid \Theta) \leq \mathsf{h}(\Theta^\mathsf{T} X) \leq k\,\mathsf{h}(\mathcal{N}(0,1)),$$

To get a lower bound on the joint entropy, we first rewrite the $(k+1)$-dimensional vector as a transformation of the $k$-dimensional $X$ and perform QR-decomposition of the $(k+1) \times d$ matrix

$$\begin{pmatrix} \Theta^\mathsf{T} X \\ X_1 \end{pmatrix} = \begin{pmatrix} \Theta^\mathsf{T} \\ e_1^\mathsf{T} \end{pmatrix} X = \begin{pmatrix} I \\ r^\mathsf{T} & \|\tilde{u}\| \end{pmatrix} \begin{pmatrix} \Theta^\mathsf{T} \\ u^\mathsf{T} \end{pmatrix} X = \mathrm{R}^\mathsf{T} \mathrm{U}^\mathsf{T} X,$$

where $u = \tilde{u}/\|\tilde{u}\|_2$ with $\tilde{u} = (I - \Theta\Theta^\mathsf{T})e_1$. Here $\mathrm{R} \in \mathbb{R}^{(k+1)\times(k+1)}$ is a full-rank upper-triangular matrix, and $\mathrm{U} \in \mathrm{St}(k+1, d)$. Then,

$$\mathsf{h}(\Theta^\mathsf{T} X, X_1 \mid \Theta) = \mathsf{h}(\mathrm{R}^\mathsf{T} \mathrm{U} X \mid \Theta) = \mathsf{h}(\mathrm{U} X \mid \Theta) + \mathbb{E}\log|\det \mathrm{R}|.$$

To lower bound the entropy in the RHS, we make use of the result from (Guo et al., 2006, Theorem 3):

$$\mathsf{h}(\mathrm{U} X \mid \Theta) \geq \mathbb{E}\,\mathrm{Tr}(\mathrm{U}\,\mathrm{diag}(\mathsf{h}(X_1), ..., \mathsf{h}(X_d))\mathrm{U}^\mathsf{T}) = E\,\mathbb{E}\,\mathrm{Tr}(\Theta\Theta^\mathsf{T} + u^\mathsf{T} u) = E\,(k+1).$$

Noting that $\log|\det \mathrm{R}| = \frac{1}{2}\log\|\tilde{u}\|_2^2 = \frac{1}{2}\log\left(1 - \left\|\Theta^\mathsf{T} e_1\right\|_2^2\right)$, the joint entropy bound is

$$\mathsf{h}(\Theta^\mathsf{T} X, X_1 \mid \Theta) \geq E\,(k+1) + \frac{1}{2}\mathbb{E}\log\left(1 - \left\|\Theta^\mathsf{T} e_1\right\|_2^2\right).$$

Since $\left\|\Theta^\mathsf{T} e_1\right\|_2^2 = \theta_{11}^2 + ... + \theta_{1k}^2 \sim \frac{Z_1^2 + ... Z_k^2}{Z_1^2 + ... Z_d^2} \sim \mathrm{Beta}\left(\frac{k}{2}, \frac{d-k}{2}\right)$ with i.i.d. $Z_i \sim \mathcal{N}(0,1)$, one concludes that

$$\mathsf{I}(\Theta^\mathsf{T} X; Y \mid \Theta) \leq k\,\mathsf{h}(\mathcal{N}(0,1)) + E - E\,(k+1) - \frac{1}{2}\left[\psi\left(\frac{d-k}{2}\right) - \psi\left(\frac{d}{2}\right)\right]$$

$$= k\,\mathsf{h}(\mathcal{N}(0,1)) - kE - \frac{1}{2}\left[\psi\left(\frac{d-k}{2}\right) - \psi\left(\frac{d}{2}\right)\right]$$

$\square$

**Lemma C.3.** Under the assumptions of Lemma C.2 holds

$$\mathsf{SI}_k(X; Y) \leq k\,\mathrm{const} - \frac{1}{2}\log\left(1 - \frac{k}{d}\right) + \frac{1}{d-k} - \frac{1}{2d},$$

where "const" is independent of $\mathsf{I}(X; Y)$.

*Proof of Lemma C.3.* By using the inequalities on the digamma function (Elezovic et al., 2000), one has the following upper bound:

$$\mathsf{SI}(X; Y) \leq k\,\mathrm{const} - \frac{1}{2}\left[\psi\left(\frac{d-k}{2}\right) - \psi\left(\frac{d}{2}\right)\right]$$

$$\leq k\,\mathrm{const} - \frac{1}{2}\left[\log\left(\frac{d-k}{2}\right) - \frac{2}{d-k} - \left[\log\left(\frac{d}{2}\right) - \frac{1}{d}\right]\right]$$

$$= k\,\mathrm{const} - \frac{1}{2}\log\left(1 - \frac{k}{d}\right) + \frac{1}{d-k} - \frac{1}{2d}.$$

$\square$

We note that both Lemma 4.1 and Lemma C.2 are much stronger than (Goldfeld and Greenewald, 2021, Proposition 1, part 2): the latter merely states that $\mathsf{SI}(X; Y) \leq \overline{\mathsf{SI}}(X; Y)$. For

instance, given the example from Lemma 4.1, (Goldfeld and Greenewald, 2021, Proposition 1, part 2) yields $\mathsf{SI}(X;Y) \leq \frac{1}{d} \mathsf{I}(X;Y)$, while our result suggests $\mathsf{SI}(X;Y) \leq \frac{1}{d-1}$, which does not depend on mutual information. Therefore, the saturation is strong (even $\mathsf{I}(X;Y) \to \infty$ does not break it) and can not be explained solely by non-optimality of projections.

## D  GAUSSIAN CHANNEL

To explore the SMI's preference for redundant over linearly extractable information, we analyze an additive white Gaussian noise (AWGN) channel. Consider a $d$-dimensional random vector $X$ with $\mathrm{cov}(X) = \mathrm{I}$, independent noise $Z \sim \mathcal{N}(0, \sigma^2\mathrm{I})$, and the channel output $Y = AX + Z$, where the matrix $A$ satisfies $\mathrm{diag}\,AA^\mathsf{T} = \mathrm{I}$ to ensure energy preservation across dimensions.

In classical information theory, maximizing $\mathsf{I}(X;Y)$ with respect to the input distribution under energy constraints $\mathbb{E}[X_i^2] = 1$ is achieved by $X \sim \mathcal{N}(0, \mathrm{I})$ (Cover and Thomas, 2006). This solution is optimal because decorrelated features provide maximal robustness against isotropic noise. When the transformation matrix $A$ is well-conditioned (i.e., $\kappa(A) \overset{\text{def}}{=} \|A\| \cdot \|A^{-1}\| \approx 1$), information about $X$ is spread evenly across the dimensions of $Y$. In contrast, as shown below, SMI exhibits the opposite preference due to its redundancy bias.

When $A = \mathrm{I}$, the channel decouples into independent scalar channels $Y_i = X_i + Z_i$. In this case, linear estimation via the conditional expectation $\frac{1}{1+\sigma^2}\mathbb{E}[X_i \mid Y_i]$ achieves the optimal mean squared error (MSE), representing the most efficient linear extraction of information.

Contrary to the theoretical optimality of well-conditioned transformations for mutual information, SMI increases with $\kappa(A)$ as shown in Figure 7. This demonstrates that SMI does not measure linearly extractable information but rather favors redundant information.

SMI's preference for ill-conditioned $A$ (high $\kappa(A)$) arises because such transformations create strong dependencies among the output features. A high condition number implies that the components of $AX$ become highly correlated, making the same information available repeatedly across different one-dimensional projections.

## E  RELATION TO OTHER MEASURES OF DEPENDENCE

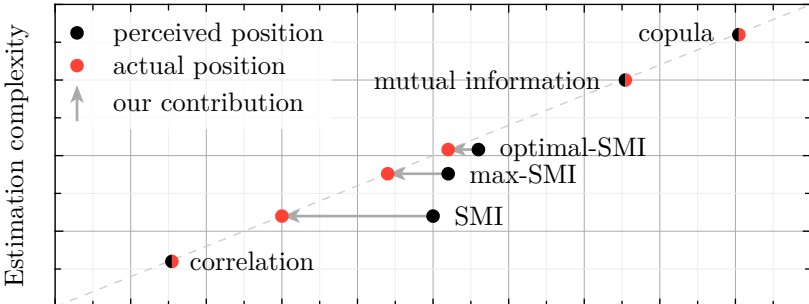

Ability to capture complex statistical dependencies

In our visual abstract, we position SMI as more complex and capable than correlation analysis but less complex than MI and copulas. In this section, we elaborate on this ranking.

- Copulas provide the most complete description of dependencies of two random vectors. The joint distribution $\mathbb{P}_{X,Y}$ fully captures probabilistic dependencies, but includes irrelevant information about marginal distributions $\mathbb{P}_X \otimes \mathbb{P}_Y$. A copula $C_{X,Y}$ factors out the former w.r.t. the latter by pinning the marginals to be uniform, thus describing the pure dependence structure (Fan and Henry, 2021). While offering full generality, copulas complexity often makes their direct use impractical.

- Mutual Information (MI) is a measure of statistical dependence, capturing non-linear relationships between two random vectors. It projects the copula onto a scalar summarizing dependence strength:

$$\mathsf{I}(X;Y) = \mathbb{E} \, \mathrm{PMI}(X;Y) = \mathbb{E} \log \frac{\mathrm{d} \, \mathbb{P}_{X,Y}}{\mathrm{d} \, \mathbb{P}_X \otimes \mathbb{P}_Y}(X,Y),$$

where the log-derivative PMI refers to Pointwise Mutual Information and literally equals to the copula $C_{X,Y}$. Thus, MI is a functional of the copula (Chen et al., 2025; Ma and Sun, 2011), and if the corresponding PDF exists, one can write

$$\mathsf{I}(X;Y) = - \, \mathsf{h}\big(C_{X,Y}\big).$$

- Sliced Mutual Information (SMI) estimates the mutual information between two random variables by averaging across one-dimensional projections. It can detect non-linear dependencies. However, as our work demonstrates, SMI saturates prematurely, prefers information redundancy, and asymptotically vanishes as the dimension growth.

- Correlation measures linear dependence. It is computationally efficient, but fails to detect any non-linear relationships.

In summary, our work shows that there exists a fundamental trade-off between computational scalability of a dependence estimator and its ability to capture rich, high-dimensional dependencies. We find that SMI, contrary to earlier assumptions, fails to overcome this trade-off. The cost of its computational benefits are misleading biases. While our findings are solid, we would like to emphasize that the visual abstract represents our personal, informal opinion, although it is backed by concrete evidence.

## F  SELECTING $k_{\mathrm{NN}}$ IN KSG ESTIMATOR

In this section, we use the same benchmarks from Section 5 to determine the optimal number of nearest neighbors ($k_{\mathrm{NN}}$) for the KSG estimator (Kraskov et al., 2004). We focus exclusively on plain Mutual Information estimation, as it is a direct component of the SMI estimation task. For each MI value from 0 to 10 in steps of 1, we perform 10 independent runs with $10^4$ samples each. We then compute the median across these runs and use it to derive the Mean Absolute Error (MAE) for different distributions and $k_{\mathrm{NN}}$ values. These errors are reported in Table 1. From the results it is evident that $k_{\mathrm{NN}} = 1$ is the best choice on average. This is consistent with Figure 4 in (Kraskov et al., 2004), where $k_{\mathrm{NN}}/N_{\mathrm{samples}} \to 0$ increases accuracy.

Table 1: MAE of the KSG estimates under different distributions and values of $k_{\mathrm{NN}}$.

|  | $k_{\mathrm{NN}}$ | | | | | |
|---|---|---|---|---|---|---|
| Distribution | 1 | 2 | 3 | 5 | 10 | 20 |
| Correlated Normal | 1.32 | 1.47 | 1.57 | 1.69 | 1.87 | 2.08 |
| Correlated Uniform | 1.45 | 1.59 | 1.68 | 1.80 | 1.98 | 2.17 |
| Smoothed Uniform | 1.42 | 1.57 | 1.67 | 1.80 | 1.98 | 2.18 |
| Log-Gamma-Exponential | 0.41 | 0.52 | 0.60 | 0.72 | 0.91 | 1.15 |

## G  ADDITIONAL EXPERIMENTS

In this section, we conduct supplementary experiments to evaluate SMI under a broader range of setups.

### G.1 Low-dimensional synthetic tests

We begin by assessing $k$-SMI on the same set of benchmarks from Section 5. The results for $k = 1, 2, 3$ are presented in Figure 4, Figure 13, and Figure 14, respectively. Notably, saturation remains consistent even for $k = d - 1$ (i.e., when only one component is discarded).

Next, we examine a setup involving randomized distribution parameters, following the methodology of (Butakov et al., n.d.). Among other adjustments, this includes randomizing per-component mutual information (e.g., assigning interactions unevenly in this experiment). In some cases (e.g., the log-gamma-exponential distribution), this increases linear redundancy, as component pairs with higher mutual information also exhibit higher variance in this particular scenario. Our results are displayed in Figure 15.

Due to numerical constraints, we do not track $I(X; Y)/d$ in this particular setup, instead plotting the results against the total mutual information. While this makes saturation slightly less evident, the general trend of SMI decreasing with $d$ remains observable. We also highlight the log-gamma-exponential distribution (Figure 15d), where SMI is less prone to saturation under parameter randomization due to the reasons mentioned earlier.

### G.2 Synthetic images

Using the MI-preserving smooth injective mappings from (Butakov et al., n.d.), we reproduce the synthetic datasets used in (Butakov et al., 2024). These datasets consist of high-dimensional images (see Figure 3) with known ground-truth mutual information. The results presented in Figure 16 again prove our findings.

### G.3 Real images with synthetic copulas

Following the technique proposed in (Lee and Rhee, 2024), we conduct additional experiments on the MNIST dataset. We consider the Markov chain:

$$X_1 \longrightarrow C_1 \rightarrow C_2 \longrightarrow X_2,$$

where $C_1$ and $C_2$ are random class variables, and $X_1, X_2$ represent random images drawn from classes $C_1$ and $C_2$, respectively. We control the mutual information $I(C_1; C_2)$ using the noisy symmetric channel framework from (Lee and Rhee, 2024). If images are selected independently given the class pair, it can be shown that $I(X_1; X_2) = I(C_1; C_2)$.

We vary $I(C_1; C_2)$ from 0 to $\log(\#\text{classes})$ (its theoretical maximum) and conduct 10 independent runs. The resulting values of $k$-SMI, averaged over 10 independent runs, are presented in Table 2. These results also indicate saturation of SMI. Moreover, one can also notice that SMI between independent is non-zero and only twice as small compared to the case $I(X_1; X_2) = 2.3$ nats, which highlights the curse of dimensionality.

Table 2: SMI results (in $10^{-3}$ nats) for the experiments with MNIST dataset.

|  | $I(X_1; X_2)$, nats | | | | | |
| --- | --- | --- | --- | --- | --- | --- |
|  | 0.0 | 0.5 | 1.0 | 1.5 | 2.0 | 2.3 |
| $SI(X_1; X_2)$, $10^{-3}$ nats | 2.89 | 2.88 | 3.80 | 4.68 | 5.73 | 5.77 |

## H Replication study: details

The original papers on $k$-SMI and max-SMI feature several experiments on independence testing and InfoMax tasks (Goldfeld et al., 2022, Section 5; Goldfeld and Greenewald, 2021, Sections 4.2,4.3; Tsur et al., 2023, Section 5). In this section, we attempt to replicate these tests to understand how their results align with our analysis.

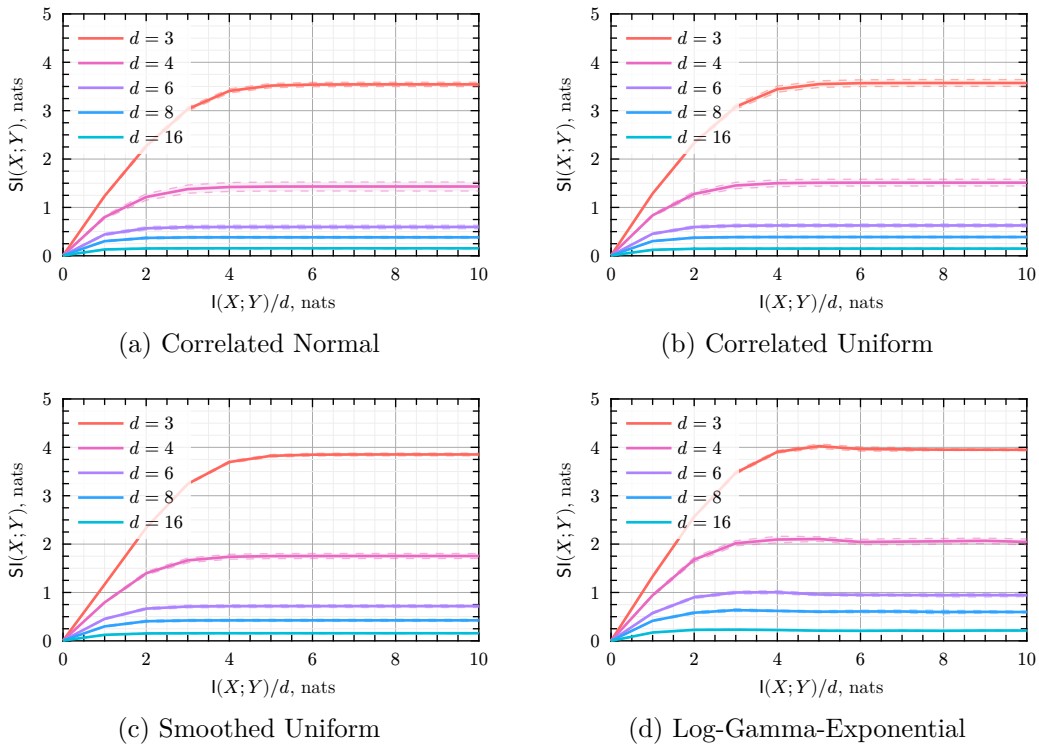

Figure 13: 2-SMI results on synthetic benchmarks. Mean values and standard deviations across 10 runs are reported, $10^4$ samples from $X, Y$ and 128 random projections were used.

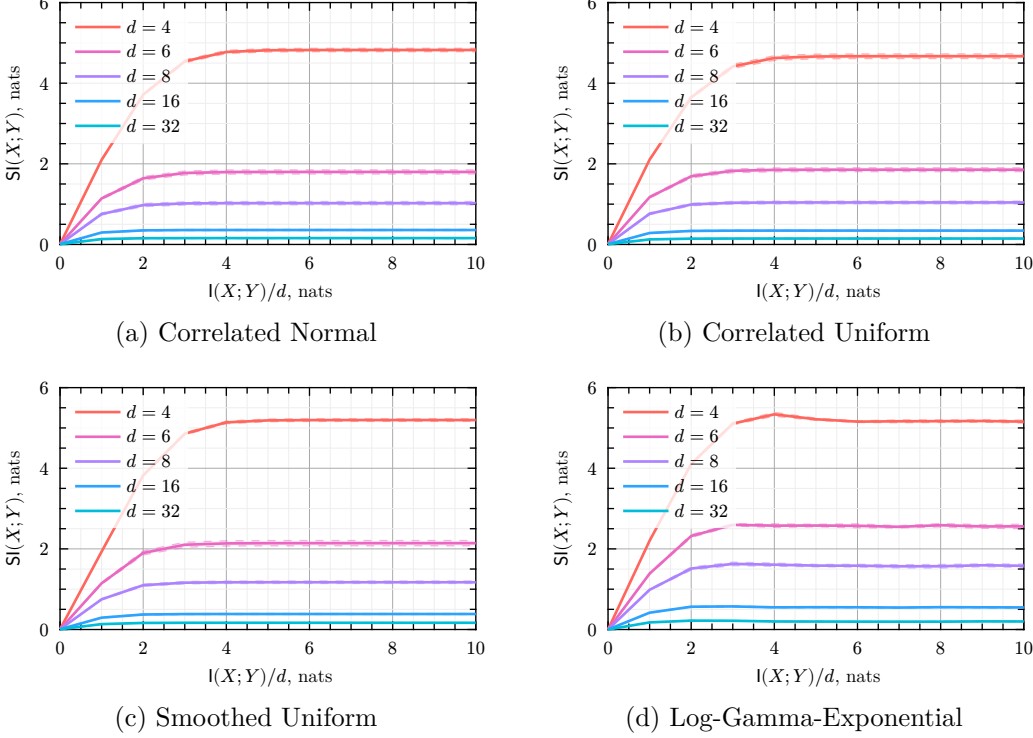

Figure 14: 3-SMI results on synthetic benchmarks. Mean values and standard deviations across 10 runs are reported, $10^4$ samples from $X, Y$ and 128 random projections were used.

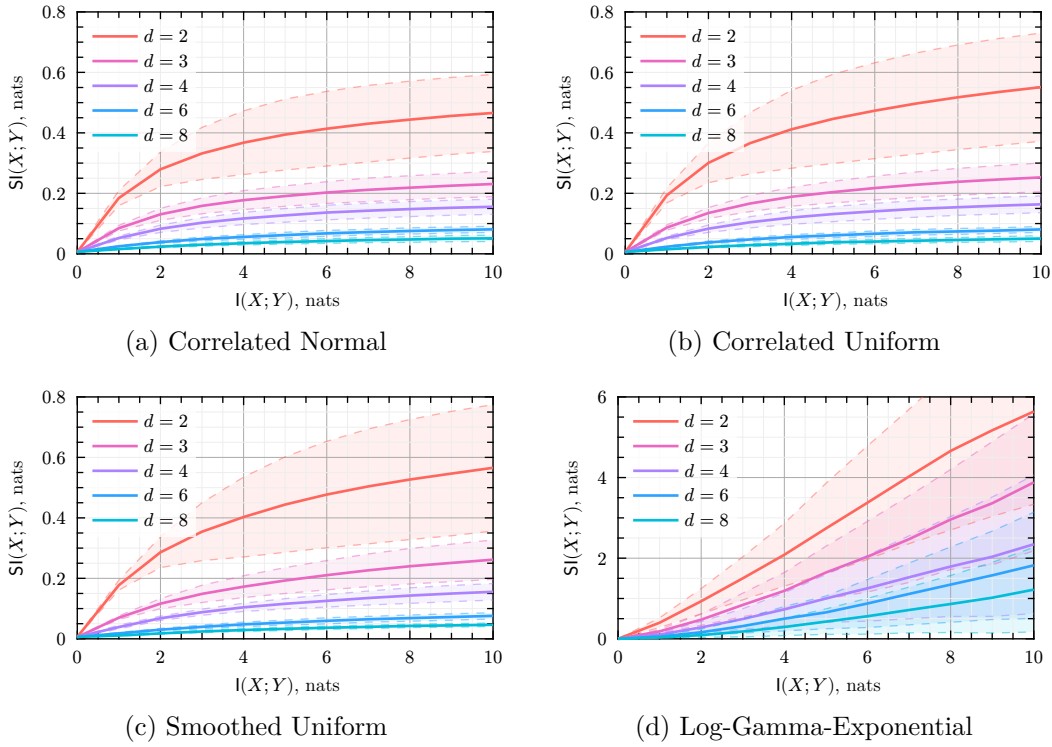

Figure 15: SMI results on synthetic benchmarks. Mean values and standard deviations across 10 runs are reported, $10^4$ samples from $X, Y$ and 128 random projections were used.

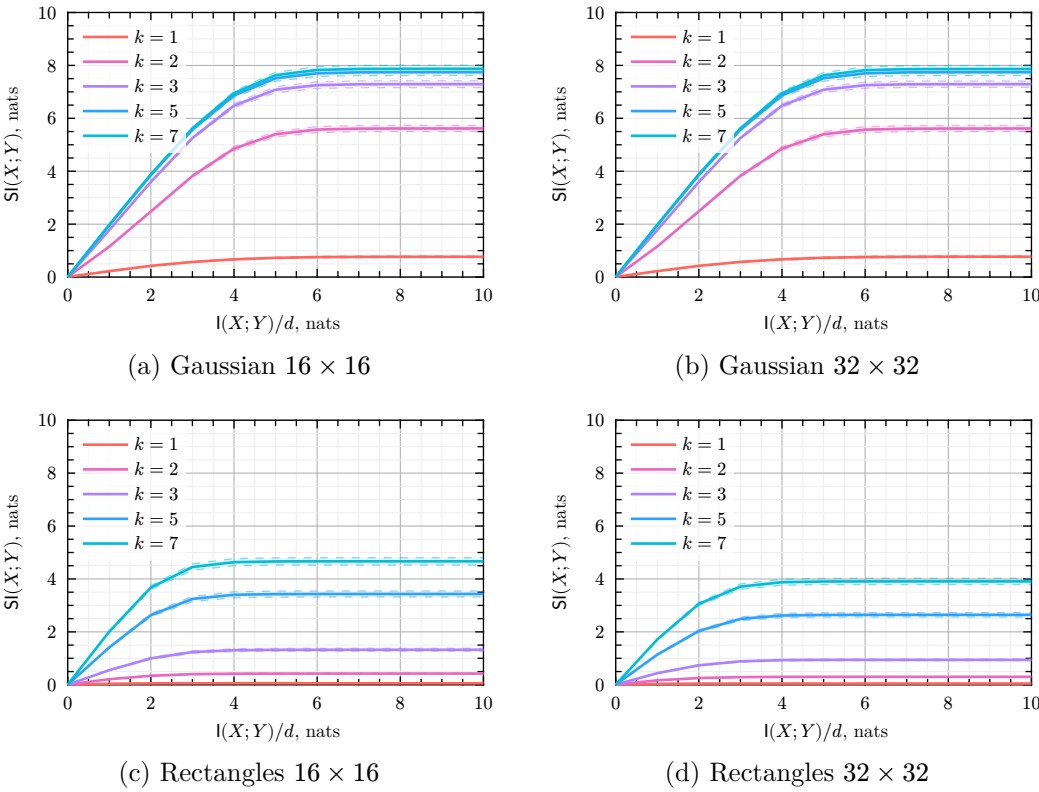

Figure 16: Results of synthetic experiments with high-dimensional image-based distributions for k-SMI. We report mean values and standard deviations computed across 10 runs, with $10^4$ samples used for MI estimation and 128 for averaging across projections.

## H.1 Feature extraction

Here, we reproduce and elaborate on the InfoMax-like feature extraction experiments. In contrast to the tasks described in Section 6 of our work, (Goldfeld and Greenewald, 2021, Section 4.3) considers a *supervised* feature extraction setting. In this setup, the shared information between $f(X)$ and $g(Y)$ is maximized with respect to the functions $f$ and $g$.

**Toy Gaussian example.** Here we consider two families of Gaussian baselines: *high redundancy* $(X, Y')$ and *low redundancy* $(X, Y'')$:

$$X, Z \sim \mathcal{N}(0, \mathrm{I}_d), \quad X \!\perp\!\!\!\perp Z \qquad Y' = Z + \sum_{i=1}^{m} \mathbf{1} \cdot e_i^\mathsf{T} X \qquad Y'' = Z + \sum_{i=1}^{m} e_i \cdot e_i^\mathsf{T} X,$$

where $\mathbf{1}^\mathsf{T} = (1, ..., 1)$ and $m$ controls the number of components that are injected into $Y$. Setting $d = 10$ and $m = 1$ for $(X, Y')$ recovers the experiment from (Goldfeld and Greenewald, 2021). However, to highlight SMI's deficiencies, we will adhere to the *low redundancy* benchmark, according to which a proper feature extraction should result in the selection of at least $m$ features.

In our experiments, we closely follow the setup from (Goldfeld and Greenewald, 2021): when maximizing $k$-SMI, we use linear $f$ and $g$, parametrized by $\mathbb{R}^{d \times d}$ matrices. However, when extracting features through MI and max-SMI maximization, we have to form a dimensionality bottleneck by using $\mathbb{R}^{k \times d}$ matrices: otherwise, the best strategy is to extract every feature. As we show below, SMI does not require this bottleneck, because it is implicitly biased toward degenerate solutions.

Similar to Section 6, variational representations are employed to conduct the experiments: the NNs from Section I are trained for 100 epochs; other settings are the same.

To evaluate the quality of the extracted features, we compute the effective rank of the matrices $\mathrm{A}_{(1:m)}, \mathrm{B}_{(1:m)}$ formed by the first $m$ columns of $\mathrm{A}$ and $\mathrm{B}$ respectively. The effective rank is defined as $\mathrm{erank}\,\mathrm{M} = \exp(\mathsf{H}(\sigma))$, where $\mathsf{H}(\sigma)$ is the Shannon entropy of the normal-

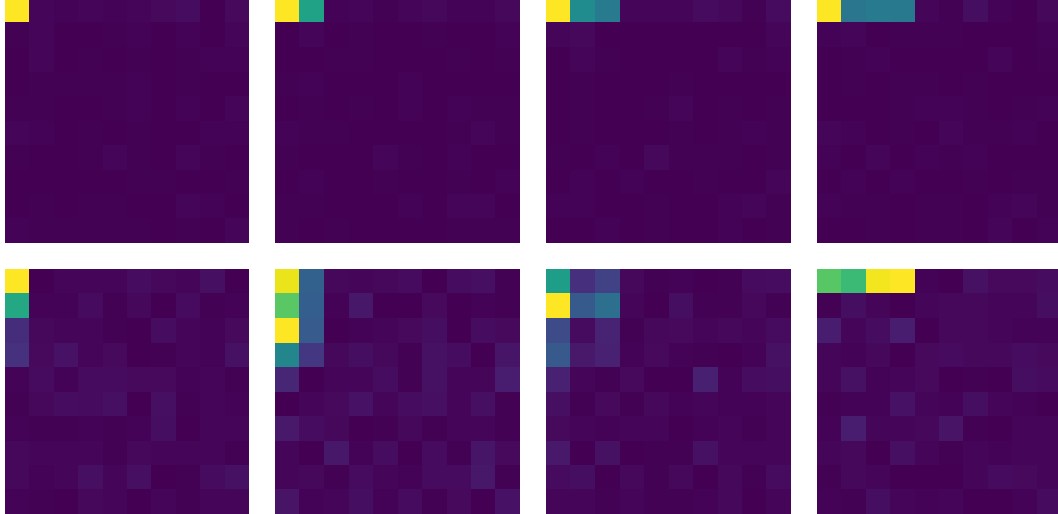

Figure 17: Feature extraction matrix for the low redundancy setting acquired through $k$-SMI $\to$ max for $m \in \{1, 2, 3, 4\}$ (columns) and $k \in \{1, 4\}$ (rows).

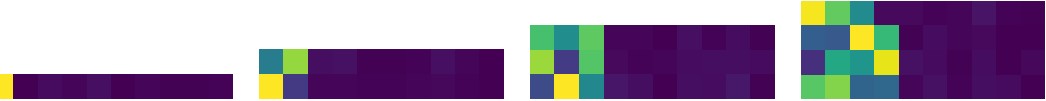

Figure 18: Feature extraction matrix for the low redundancy setting acquired through MI $\to$ max for $m \in \{1, ..., 4\}$ (rows).

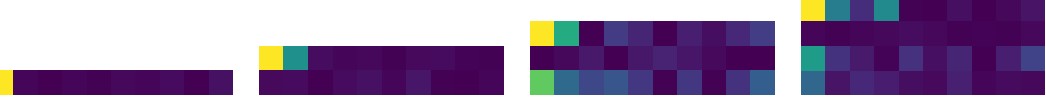

Figure 19: Feature extraction matrix for the low redundancy setting acquired through 1-max-SMI → max for $m \in \{1, ..., 4\}$ (rows).

ized singular values $\sigma$ of M. If $\text{erank } A_{(1:m)} \approx m$, then all features are extracted without mixing, while low values of $\text{erank } A_{(1:m)}$ indicate the (numerically) irrecoverable collapse to mixtures (ill-posed linear combinations of the first $m$ components of $X$).

Our results, depicted in Figure 9, show that the MI maximization yields effective rank close to $m$, confirming its ability to recover all relevant features. In contrast, $k$-SMI yields an effective rank that nearly constant regardless of $k$, revealing its redundancy bias. This collapse confirms that $k$-SMI optimization leads to redundant features.

## H.2 INDEPENDENCE TESTING

The SMI has been proposed as a scalable alternative to MI for independence testing (Goldfeld et al., 2022; Goldfeld and Greenewald, 2021; Nuradha and Goldfeld, 2023; Tsur et al., 2023), which can be framed as a binary classification task. Given estimates of SMI (or MI) on datasets drawn from either the joint distribution (positive class) or the product of marginals (negative class, obtained by shuffling), one can apply the threshold for dependence verification. For each fixed dimension $d$, and sample size $n$, we can generate 100 positive and 100 negative pairs of samples, estimate SMI (or MI), and compute the ROC-AUC over these 200 scored examples as a function of the number of samples $n$. The works (Goldfeld et al., 2022; Goldfeld and Greenewald, 2021) show that SMI outperforms MI when the dimension is fixed.

We replicate this protocol with one critical modification. We pool estimates across different dimensions ($d \in \{2, 10, 20, 30\}$) for each sample size $n$, and then compute a single ROC-AUC from the mixed-dimensional data. Additionally, we fix the ground truth MI to 1 and 2 nat for each dataset and replace the Kozachenko–Leonenko estimator used in (Goldfeld and Greenewald, 2021) with the KSG estimator (Kraskov et al., 2004) (using $k_{\text{NN}} = 1$ neighbors), which in our experiments yields more stable MI estimates.[1] For a fair comparison we report MI values over 128 random rotations, because it showed numerically improved MI estimates for small-size datasets.

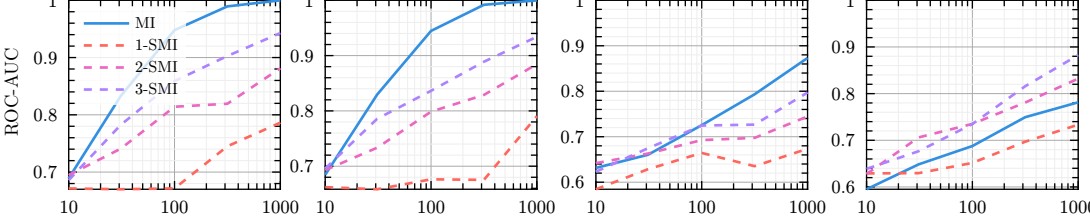

Figure 20: Independence testing: ROC-AUC versus sample size for correlated normal, correlated uniform, smoothed uniform and log-gammma-exponential (left-to-right, 1 nat).

As shown in Figures 11, 20, this pooling causes SMI's discriminative power to drop sharply, while MI's remains high. The failure occurs because SMI decays with dimension even when total mutual information is held constant, so dependent high-dimensional cases produce SMI values that overlap with independent low-dimensional cases. The slower dimensional decay of SMI for LGE distribution (Figures 5, 15), in turn, explain the observed higher ROC-AUC. Consequently, SMI is less reliable for independence testing than MI unless the

---

[1]By using the KSG estimator, we observe that ROC-AUC dynamics corresponding to MI come into closer agreement with those of SMI, which is not seen when using the less stable Kozachenko–Leonenko estimator.

dimensionality is known and fixed in advance, which imposes a strict limitation for practical applications where data dimensionality may vary.

# I    IMPLEMENTATION DETAILS

## I.1    SYNTHETIC EXPERIMENTS

For the experiments from Section 5, we use implementation of Kraskov-Stoegbauer-Grassberger (KSG) (Kraskov et al., 2004) mutual information estimator and random slicing from (Butakov et al., n.d.). The number of neighbors is set to $k_{\mathrm{NN}} = 1$ for the KSG estimator. For each configuration, we conduct 10 independent runs with different random seeds to compute means and standard deviations. Our experiments use $10^4$ samples for $(X, Y)$ and 128 samples for $(\Theta, \Phi)$.

For the experiments from Section 5, we use independent components with equally distributed per-component MI. For the supplementary experiments from Figure 15, parameters of each distribution (e.g., covariance matrices) are randomized via the algorithm implemented in (Butakov et al., n.d.). This includes randomization of per-component MI (which is done using a uniform distribution over a $(d-1)$-dimensional simplex).

For the experiments, we used AMD EPYC 7543 CPU, one core per distribution. Each experiment (fixed $k$, varying $d$) took no longer then 3 days to compute.

## I.2    REPRESENTATION LEARNING EXPERIMENTS

Recall that Deep InfoMax requires maximizing a lower bound on $\mathsf{I}(X; f(X))$, where $X$ is input data and $f$ is an encoder network. Since $\mathsf{I}(X; f(X))$ is typically vacuous, the lower bound in question should be selected carefully to (a) be finite and (b) allow for meaningful optima. In our experiments, we employ the objective from (Butakov et al., 2025), which provably satisfies the requirements above, while also being inherently regularized against representation collapse:

$$I(f(X'); f(X) + Z) \leq \mathsf{I}(X; f(X)),$$

where $Z$ is Gaussian and independent, and $X'$ represents randomly augmented data.

For experiments on MNIST dataset, we use a simple ConvNet with three convolutional and two fully connected layers. A three-layer fully-connected perceptron serves as a critic network for the InfoNCE loss. We use the same architecture and loss for SMI maximization. As described in (Goldfeld et al., 2022; Goldfeld and Greenewald, 2021), the critic network for the SMI lower bound takes $\Theta^\mathsf{T} X$, $\Phi^\mathsf{T} Y$, $\Theta$ and $\Phi$ as inputs. To accommodate the flattened $\Theta$ and $\Phi$ matrices, we increase the network's input dimensionality; the rest of the architecture remains unchanged. The details are provided in Table 3. When maximizing SMI, we generate a set of random projectors for each batch of samples from $X, Y$, with one projector per sample.

We use additive Gaussian noise with $\sigma = 0.2$ as an input augmentation. Training hyperparameters are as follows: batch size $= 512$, 2000 epochs, Adam optimizer (Kingma and Ba, 2017) with learning rate $10^{-3}$.

For the experiments, we used AMD EPYC 7543 CPU and Nvidia A100 GPUs. Each experiment took no longer then 1 day to compute.

Table 3: The NN architectures used to conduct the tests on MNIST images in Section 6.

| NN | Architecture |
|---|---|
| ConvNet, $24 \times 24$ images | $\times 1$: Conv2d(1, 32, ks=3), MaxPool2d(2), BatchNorm2d, LeakyReLU(0.01) 
 $\times 1$: Conv2d(32, 64, ks=3), MaxPool2d(2), BatchNorm2d, LeakyReLU(0.01) 
 $\times 1$: Conv2d(64, 128, ks=3), MaxPool2d(2), BatchNorm2d, LeakyReLU(0.01) 
 $\times 1$: Dense(128, 128), LeakyReLU(0.01), Dense(128, dim) |
| Critic NN for MI, pairs of vectors | $\times 1$: Dense($2 \times$ dim, 256), LeakyReLU(0.01) 
 $\times 1$: Dense(256, 256), LeakyReLU(0.01), Dense(256, 1) |
| Critic NN for SMI, pairs of vectors | $\times 1$: Dense($2 \times$ k $+ 2 \times$ dim $\times k$, 256), LeakyReLU(0.01) 
 $\times 1$: Dense(256, 256), LeakyReLU(0.01), Dense(256, 1) |

