# OpenReview forum: "Curse of Slicing: Why Sliced Mutual Information is a Deceptive Measure of Statistical Dependence"
_ICLR.cc/2026/Conference — ICLR 2026 Poster_

### Official Review · Reviewer_pnxw · 2025-10-16

**Soundness:** 3
**Presentation:** 3
**Contribution:** 3
**Rating:** 6
**Confidence:** 3

**Summary:**

Theoretical properties of sliced mutual information (SMI) are studied, suggesting strong limitations of SMI in application scenarios in which it has been employed in the past, e.g., the Deep InfoMax setting. Furter, discrepancies between SMI and conventional mutual information are illustrated, that is, SMI between deterministic Gaussian variables is bounded while MI is infinite. For increasing dimensionality SMI approaches zero, while again, MI is approaching infinity. The theoretical analysis has been supported with simulation experiments.

**Strengths:**

- The paper highlights severe limitations of SMI that are relevant for practitioners.
- Theoretical results for multivariate Gaussian variables have been derived which illustrate the counterintuitive behaviour of SMI compared to MI.
- The main theoretical result derived in Lemma 4.1 has been supported with sufficient experimental evidence.
- The usecase of Deep InfoMax has been investigated at the example of a small image dataset.

**Weaknesses:**

While the discrepancy between MI and SMI is well-illustrated, the claims regarding the Deep InfoMax principle are not as strongly supported by the experiments:
- The description provided in D.2 explains how to obtain the baseline but more details regarding the implementation of SMI are needed.
- The results are only obtained for a relatively small image dataset (MNIST). To support the strong impact statement made by the authors, more complex datasets such as CIFAR10, or similar ones should be analyzed.

 Some experimental details are missing:
- How exactly was $SI_k$ implemented for the Deep InfoMax experiment?
- How many seeds did the authors consider to confirm their results?
- Why is $k=1$ for the KSG estimator? Typically a higher $k$ such as 5 or 7 is more stable.

Minor:
- In their discussion of the Deep InfoMax experiments, the authors should mention that, e.g., for invertible $f$ we get that $I(X;f(X))$ is infinite and we can only compute it if $f$ fulfills certain properties that avoid this behaviour.
- Punctuation missing in Lemma A.3.

**Questions:**

Questions:
- Are there any usecases for which the authors would recommend to use SMI instead of MI?
- Is the proof of Lemma A.4 just recited from 41, or are there any new contributions in it?
- Do the works that used SMI for Deep InfoMax [22,23] employ any architectural restrictions or similar to avoid collapse?

---

> ### Author Response · Authors · 2025-11-21
> **Official Comment**
>
> Dear Reviewer **pnxw**,
>
> Thank you for your valuable feedback and insightful comments! In the following text, we provide responses to your concerns. We hope that all of them are addressed properly.
>
> **Weaknesses:**
>
> 1. > ... the claims regarding the Deep InfoMax principle are not as strongly supported by the experiments: ... more details regarding the implementation of SMI are needed.
>
>    Thank you for pointing this out. We address this concern directly in our answer further below. The corresponding details are now also clarified in the manuscript.
>
> 1. > The results are only obtained for a relatively small image dataset (MNIST). ... more complex datasets such as CIFAR10, or similar ones should be analyzed.
>
>    We performed a controlled validation of SMI's properties by precisely reproducing the experimental setup from [r.1], which has already demonstrated its efficacy across MNIST, CIFAR, and ImageNet when maximizing MI lower bounds. Within this validated setup, simply replacing MI with SMI was sufficient to cause a clear representation collapse on MNIST (see Section 6). We are confident that this failure on a standard benchmark provides conclusive evidence of SMI's fundamental limitations. Therefore, we consider additional experiments on more complex datasets to be superfluous to our paper's core argument.
>
>    Nevertheless, to be thorough, we are conducting similar Deep SMI-Max experiments on CIFAR-10. Preliminary results again indicate an immediate collapse. After completing more independent trials, we will include the corresponding results in the Appendix. We maintain, however, that this information is supplementary, as **SMI already fails in simpler setups**.
>
> 1. > How exactly was SI_k implemented for the Deep InfoMax experiment?
>
>    We use the same network as for plain Deep InfoMax but provide $\Theta$ and $\Phi$ as additional inputs. This approach strictly follows the implementation of neural $k$-SMI estimator from the original papers that introduce SMI. We have clarified this point in Appendix H.2 of the revised manuscript.
>
> 1. > Why is k=1 for the KSG estimator? Typically a higher k such as 5 or 7 is more stable.
>
>    Our study shows that $k\_\text{NN}=1$ provides the best quality of estimates. Please, visit Appendix F of the revised manuscript for further details and grid search results.
>
> 1. > ... the authors should mention that, e.g., for invertible f we get that I(X; f(X)) is infinite ...
>
>    We thank the reviewer for this important remark. As stated in line 398, we optimize a *lower bound* of $I(X; f(X))$ derived in [r.1] (namely, $I(f(X'); f(X) + Z)$, where X' is an augmentation of X and Z is an independent noise; this is now mentioned in the current revision). This lower bound is finite by construction due to the injected noise Z, which resolves the potential problem of infinite MI for deterministic, invertible $f$. These clarifications are now added to Appendix H.2 of the revised manuscript, including the remark about infinite MI.
>
> 1. > Punctuation missing in Lemma A.3.
>
>    Thank you! This has been fixed in the newer revision.
>
> **Questions:**
>
> 1. > Are there any usecases for which the authors would recommend to use SMI instead of MI?
>
>    SMI should be applied with caution. It may still have redeemable applications as a scalable measure in settings where one is primarily interested in detecting the presence of any dependence between vectors of moderate dimensionality, and not in its strength or quality. However, the core issue is that any observed correlation between SMI and a metric of interest can be misleading, as SMI's dynamics are non-trivial and often counter-intuitive (for example, it can behave conversely to vanilla MI, as we show in our work).
>
> 1. > Is the proof of Lemma A.4 just recited from 41, or are there any new contributions in it?
>
>    The proof of Lemma A.4 is just recapitulated from [41, Theorem 1.5 and Proposition 1.2] to make our presentation self-contained. We also have adapted it to our specific notations and context.
>
> 1. > Do the works that used SMI for Deep InfoMax [22,23] employ any architectural restrictions or similar to avoid collapse?
>
>    Revisiting the original works, we do not notice any regularization against collapses. However, at least some of the original experiments seem to be designed to inherently favor SMI. For instance, in the feature selection task [22, Section 4.3], the 10-dimensional $Y$ is generated by adding the same source feature $X_1$​ to every dimension of an independent noise vector. This explicitly builds high redundancy into the system (as the information about $X_1$ is repeated across $10$ axes), which SMI favours.
>
> We thank Reviewer **pnxw** once again for their work! We hope our response and revisions addresses your comments clearly, and we are happy to provide further clarification if needed.
>
> [r.1] I. Butakov et al., "Efficient Distribution Matching of Representations via Noise-Injected Deep InfoMax", Proc. of ICLR 2025

---

> > ### Comment · Reviewer_pnxw · 2025-11-24
> >
> > I would like to thank the authors for their clarifications and additional experiments. Since my main concerns have been addressed, I increased my score.

---

### Official Review · Reviewer_ZesV · 2025-10-22

**Soundness:** 4
**Presentation:** 3
**Contribution:** 3
**Rating:** 8
**Confidence:** 3

**Summary:**

This paper presents an analytical and empirical critique of Sliced Mutual Information (SMI), a popular scalable alternative to mutual information (MI) used in various works on high-dimensional statistical dependence estimation and deep learning analysis. The authors argue that despite its recent adoption, SMI exhibits several fundamental flaws that make it unreliable as a measure of dependence, including the following:
1) SMI rapidly saturates even for simple synthetic problems, failing to reflect true increases in dependence.
2) SMI prioritizes redundant or repeated information rather than informative content.
3) Although promoted as dimension-robust, SMI actually decays to zero asymptotically in high dimensions.
4) SMI can increase under deterministic mappings, unlike MI.

Through theoretical analysis and experiments with synthetic data, the paper demonstrates these deficiencies and shows that using SMI as a replacement for MI is, at the very least, problematic.

**Strengths:**

Thank you for the paper.  It was an interesting read.

The paper has both detailed theoretical analysis with several key examples (such as closed-form analysis for certain classes of Gaussian variables) and a large number of synthetic experiments to validate their theoretical findings and conceptual contribution.

The paper provides an appropriately critical assessment, in a timely and important way.

The paper is well written and nicely organized.

**Weaknesses:**

The following weaknesses are suggested, but they aren't (in my mind) especially significant.

The claim that SMI fails to detect increases in dependence even for linear transformations that enhance information extraction should be clarified to have been shown for specific cases, not necessarily universally.

A specific real-world example would be interesting.

The math in this paper is detailed/hard and probably beyond many readers, but not sure that can be helped.

**Questions:**

Less questions, more summarizing comments:

My preliminary assessment is that this a somewhat niche topic (papers on “sliced mutual information” seem fairly sparse, according to Google scholar), and so I might be less inclined to accept the paper.  However, some of the papers have appeared in prominent conferences (such as NeurIPS), and as such it seems like an important outcome to have the full story available for other researchers to examine.

Perhaps my main question is, given the paper's results, whether the authors feel that SMI currently has any applications settings where it would still seem redeemable, or if it is back to the drawing board. Though that is more of a personal interest question -- I'm not sure I would want to suggest the authors should be required to make a strong statement in the paper, when I think their analysis lays out concrete issues clearly.

---

> ### Author Response · Authors · 2025-11-21
> **Official Comment**
>
> Dear Reviewer **ZesV**,
>
> We are grateful for the thorough review and valuable comments. We are also happy to hear that you enjoyed reading our work! Our responses below aim to address all the raised concerns.
>
> **Weaknesses**
> 1. > The claim that SMI fails to detect increases in dependence even for linear transformations that enhance information extraction should be clarified to have been shown for specific cases, not necessarily universally.
>
>     Thank you for your suggestion. To clarify, we demonstrate this limitation of SMI with a specific example involving a Gaussian channel with linear preconditioning: $X \to Y = \mathrm{A} X + \mathcal{N}(0, \sigma^2 I)$
>
>     In this setup, the matrix $A$ can either correlate or decorrelate the features of $X$. Since the noise is isotropic, decorrelating the features of $X$ makes them easier to reconstruct from the noisy signal $Y$, which increases the mutual information $I(X;Y)$. However, SMI increases when $\mathrm{A}$ correlates features, even though this makes them more susceptible to noise and harder to recover from $Y$. This illustrates a case where an increase in SMI does not correspond to an improvement in usable information.
>
>     The results of this experiment are detailed in the revised manuscript on lines 446-452 and in Figure 7.
>
> 2. > A specific real-world example would be interesting.
>
>     We agree with this point. Providing such an example is our next priority, and we hope to achieve it in a couple of days.
>
> 3. > The math in this paper is detailed/hard and probably beyond many readers, but not sure that can be helped.
>
>     Thank you for this feedback. We agree that the mathematical sections are demanding. We tried our best to make the manuscript more accessible by complementing the math with intuitive explanations and examples.
>
>
> **Questions**
> 1. > My preliminary assessment is that this a somewhat niche topic (papers on “sliced mutual information” seem fairly sparse, according to Google scholar)...
>
>     We agree that SMI's adoption is more limited than standard MI's. However, the limitations of MI are well-documented in the literature. For instance, foundational works highlight the exponential sample complexity of MI estimation [r.1-2] and its vacuousness for deterministic relations [r.3]. In contrast, there is a significant gap in understanding the inherent limitations of SMI, which our work aims to address.
>
>     Furthermore, we note that since this article was in progress, another paper employing SMI has been accepted to a CORE-A conference [r.4], and a new Python package for pointwise SMI estimation ([`psmi`](https://pypi.org/project/psmi/)) has been submitted to PyPI.
>
>     [r.1] Z. Goldfeld et al., "Convergence of Smoothed Empirical Measures with Applications to Entropy Estimation", ICSEE-2018, ISIT-2019
>
>     [r.2] D. McAllester, K. Stratos, "Formal Limitations on the Measurement of Mutual Information." Proc. of  AISTATS 2020
>
>     [r.3] R. A. Amjad, B. C. Geiger, "Learning Representations for Neural Network-Based Classification Using the Information Bottleneck Principle." IEEE TPAMI 2018
>
>     [r.4] J. Dentan et al., "Predicting Memorization Within Large Language Models Fine-Tuned for Classification." Proc. of ECAI 2025
>
> 2. > ...given the paper's results, whether the authors feel that SMI currently has any applications settings where it would still seem redeemable, or if it is back to the drawing board...
>
>     We think it is not entirely back to the drawing board, but rather that SMI should be applied with caution. It may still have redeemable applications as a scalable measure in settings where one is primarily interested in detecting the presence of any dependence between vectors of moderate dimensionality, and not in its strength or quality. However, the core issue is that any observed correlation between SMI and a metric of interest can be misleading, as SMI's dynamics are non-trivial and often counter-intuitive (for example, it can behave conversely to vanilla MI, as we show in our work).
>
> We once again thank Reviewer **ZesV** for their time and insightful comments! Should any further points arise, we look forward to addressing them.

---

> > ### Comment · Reviewer_ZesV · 2025-11-25
> >
> > I have seen the responses given by the authors to my and other reviews.  I appreciate their work.  I am planning to keep my score the same.

---

### Official Review · Reviewer_KQcg · 2025-10-24

**Soundness:** 2
**Presentation:** 3
**Contribution:** 2
**Rating:** 4
**Confidence:** 4

**Summary:**

This paper discusses the shortcomings of Sliced Mutual Information (SMI) as a tool for measuring statistical dependence in high-dimensional settings. The authors show that SMI can saturate as the correlation between random vectors increases and decreases asymptotically as dimension grows. The authors validate the theoretical findings with synthetic experiments.

**Strengths:**

- The paper is well-written
- The paper presents mathematical proofs of SMI’s limitations in the simple Gaussian scenario
- The paper highlights major flaws of a novel dependence measure

**Weaknesses:**

- The theoretical results are limited to a Gaussian setting (Lemma 4.1). This does not prove anything for more complex scenarios
- Since the theoretical part is limited to the Gaussian setting, one would expect the experimental results to be comprehensive of significantly complex scenarios for which the authors did not provide a theoretical contribution, to show that the paper contributions hold true for many possible cases. However, the experimental results could be improved

**Questions:**

- In the numerical experiments, are you considering a finite-data regime or infinite-data regime? I did not understand it from the paper, but it is well-known that, for instance, MI estimators perform differently depending on the regime. In any case, how does SMI perform in the other regime that you did not consider?
- How do you explain the saturation phenomenon of SMI for non-Gaussian scenarios?
- Since you compare correlation coefficients, SMI, MI, and copula in the initial figure of the paper, why did you not include any other observation or comment on these measures? Can you provide a paragraph of comparison?
- It appears that high dimensions pose a theoretical problem with SMI in the Gaussian case. However, SMI was used specifically for high-dimensionality problems. So, could it be that SMI does not have the same saturation problem for high dimensions in different scenarios? Did you find any settings in which the dimensionality was not a problem for SMI, and that are not reported in the paper?
- What does << mean in your case (line 114)?

---

> ### Author Response · Authors · 2025-11-21
> **Official Comment**
>
> Dear Reviewer **KQcg**,
>
> We sincerely thank you for your profound review. Below, we provide answers to the questions and concerns you raised.
>
> **Weaknesses:**
>
> 1. > The theoretical results are limited to a Gaussian setting (Lemma 4.1). This does not prove anything for more complex scenarios
>
>    The Gaussian case is one of the simplest in statistical analysis, as it is fully described by mean values and correlations. In this case, SMI performs even worse than plain correlation. Moreover, Lemma 4.1 illustrates that SMI's limitations are not due to a finite number of samples and projections, or inaccuracy of a backbone MI estimator. Therefore, if SMI fails here, its performance is even more dubious in more complex scenarios.
>
>    However, motivated by your valuable feedback, we have derived a more general result, which is included in Appendix C of our next revision. Specifically, we prove that if $X$ is continuous and has independent components, and the redundancy is low, $SI(X;Y)$ still saturates. For further details, please see Appendix C of the revised manuscript.
>
> 1. > ... the experimental results could be improved
>
>    The number of known unique distributions (up to diffeomorphism) with tractable mutual information is very low. Our paper features the most popular choices from several benchmarks [17,34,35,36]. We also evaluate SMI on more complex distributions with tractable MI that mimic real data in terms of both dimensionality and latent structure (Figure 3).
>
>    Nonetheless, we have updated our "Additional Experiments" section (Appendix G in the revised manuscript) to include experiments with real data, conducted according to the method proposed in [r.1]. Specifically, we sample a pair of classes $C_1,C_2$ with known ground-truth MI $I(C_1;C_2)$, and then independently sample two images $X_1,X_2$ of classes $C_1$ and $C_2$, respectively, from the MNIST dataset (Using the CIFAR-10 dataset resulted in `nan` values for SMI, a problem we are still investigating). It can be shown that $I(X_1;X_2) = I(C_1;C_2)$. Applying SMI yields results consistent with our findings: the estimated values are small and saturate immediately. For more information, please see the revised Appendix G.3.
>
>    If the reviewer has other experiments with tractable mutual information in mind, we kindly ask to share them before the discussion ends.
>
> **Questions:**
>
> 1. > ...are you considering a finite-data regime or infinite-data regime? ... how does SMI perform in the other regime ...?
>
>    For the numerical experiments, we use $10^4$ samples, as stated on lines 336 and 377. Our 10-seed confidence intervals suggest that the plots will not change significantly if the number of samples is infinite. Currently, we are unaware of methods that allow for actual infinite-data numerical experiments with our particular estimator.
>
> 1. > How do you explain the saturation phenomenon of SMI for non-Gaussian scenarios?
>
>    Informally, high MI typically corresponds to almost determenistic relation between between $X$ and $Y$, meaning that $(X,Y)$ lies close to some manifold. For us, it seems that linear projections are too crude to capture $(X,Y)$ further concentrating near this manifold.
>
> 1. > Since you compare correlation coefficients, SMI, MI, and copula in the initial figure of the paper, why did you not include any other observation or comment on these measures? Can you provide a paragraph of comparison?
>
>    We thank the reviewer for this valuable suggestion. In direct repsonse to your comment, we have included a new subsection in the appendix (Section E: Relation to other measures of dependence). We provide a comparison between correlation coefficient, SMI, MI, copulas, and discuss their respective position in the hierarchy of dependence measures presented in the initial figure.
>
> 1. > ... could it be that SMI does not have the same saturation problem for high dimensions in different scenarios? Did you find any settings in which the dimensionality was not a problem for SMI, and that are not reported in the paper?
>
>    Thank you for this question. At the time of writing the paper, we were not aware of any such distribution. However, we can now present an example. Consider any discrete (X,Y) embeded in $\mathbb{R}^{d_X + d_Y}$. One can show that almost any random projection is injective on a discrete support. Therefore, due to the invariance of MI under measurable injective transformations, $SI_k(X;Y) = I(X;Y)$. For further details, please see Appendix C in the revised manuscript.
>
> 1. > What does << mean in your case (line 114)?
>
>    By $\ll$ we denote absolute continuity of $\mathbb{Q}$ with respect to $\mathbb{P}$. We added the explanation on line 114.
>
> We would like to once again sincerely thank Reviewer **KQcg** for their work. If any concerns remain, we will be happy to address them as well.
>
> [r.1] Kyungeun Lee and Wonjong Rhee, "A Benchmark Suite for Evaluating Neural Mutual Information Estimators on Unstructured Datasets." Proc. of NeurIPS 2024

---

> > ### Comment · Reviewer_KQcg · 2025-11-24
> >
> > Thank you for your answers, which satisfied my concerns. I have raised my score.

---

### Official Review · Reviewer_AoMn · 2025-10-29

**Soundness:** 4
**Presentation:** 4
**Contribution:** 3
**Rating:** 8
**Confidence:** 4

**Summary:**

This paper gives a critical analysis of a cheap and intuitive proxy measure for statistical dependence called "sliced mutual information" (SMI).
Although the simplicity of SMI has fueled its growing popularity, this study finds that the name is deceptive and that SMI has a number of drawbacks that make it inappropriate as a proxy for statistical dependence measures like Mutual Information: SMI saturates as dependence grows, SMI is biased towards redundancy, SMI decays to zero in high dimensions. The paper builds analytic examples and uses synthetic data to demonstrate the issues, and shows that simple workarounds do not fix the problem.

**Strengths:**

- The paper was well structured. SMI and its defects were clearly described, with intuitive experiments supporting each result.

- While most of the arguments were supported by analytical counter-examples, I was happy to also see comparisons with recent synthetic data benchmarks used in the MI estimation literature.

- I appreciate that the paper went beyond showing examples where SMI gives counter-intuitive results, and also showed that *optimizing* SMI leads to poor results.

- The deficiencies of SMI (like redundancy bias) were not very surprising to me, but the curse of dimensionality effect was quite a bit stronger than I expected. (I assumed that as dimensionality grows, it would be hard to find a "good slice", but it seems that even analytically integrating slices leads to decaying SMI!)

**Weaknesses:**

I thought the critique was straightforward and clear. The only question in my mind is the broader significance of these results. I am familiar with recent neural MI estimation literature, and I confess I had not seen any mention of SMI (though I think I recall a reference to it) so I was a little surprised to see it described as a "popular" approach. Nevertheless, the citations don't lie, and a decent number of papers in top venues are studying something that is (as I am convinced by this paper) a dubious measure. Therefore, even though I don't think this has the broadest significance for the field, the critique should be at least as visible as the flawed approach.

**Questions:**

- I didn't go back to study the IB paper that you cited that optimized SMI. Your result suggests that it shouldn't work, is there any explanation on how they were able to show reasonable results to publish with this method?

- One small improvement for a reader like myself would be to situate the SMI literature with respect to the neural MI estimation literature. I just assumed that it was clear that neural MI estimation methods, like neural everything else, was the clear winner. Are there applications / properties / uses cases which led people to prefer SMI over neural methods? I assume it's just for computational simplicity?

- One other thought that might extend the impact of this work is to discuss whether there are connections to other "sliced" estimators, like sliced score matching. Even the popular Hutchinson trace estimator could be considered a sliced estimator.

---

> ### Author Response · Authors · 2025-11-21
> **Official Comment**
>
> Dear Reviewer **AoMn**,
>
> Thank you sincerely for reviewing our article! In the following text, we provide responses to your concerns. We hope that all of them are addressed properly.
>
> **Weaknesses**
> 1. > I thought the critique was straightforward and clear. The only question in my mind is the broader significance of these results...
>
>     We agree that SMI's adoption is more limited than standard MI's. However, the limitations of MI are well-documented in the literature. For instance, foundational works highlight the exponential sample complexity of MI estimation [r.1-2] and its vacuousness for deterministic relations [r.3]. In contrast, there is a significant gap in understanding the inherent limitations of SMI, which our work aims to address.
>
>       Furthermore, we note that since this article was in progress, another paper employing SMI has been accepted to a CORE-A conference [r.4], and a new Python package for pointwise SMI estimation ([`psmi`](https://pypi.org/project/psmi/)) has been submitted to PyPI.
>
>     [r.1] Z. Goldfeld et al., "Convergence of Smoothed Empirical Measures with Applications to Entropy Estimation", ICSEE-2018, ISIT-2019
>
>      [r.2] D. McAllester, K. Stratos, "Formal Limitations on the Measurement of Mutual Information." Proc. of  AISTATS 2020
>
>     [r.3] R. A. Amjad, B. C. Geiger, "Learning Representations for Neural Network-Based Classification Using the Information Bottleneck Principle." IEEE TPAMI 2018
>
>     [r.4] J. Dentan et al., "Predicting Memorization Within Large Language Models Fine-Tuned for Classification." Proc. of ECAI 2025"
>
> **Questions**
> 1. > I didn't go back to study the IB paper that you cited that optimized SMI. Your result suggests that it shouldn't work, is there any explanation on how they were able to show reasonable results to publish with this method?
>
>     That is a good question. We believe the discrepancy arises because the original experiments were designed in a way that inherently favors SMI. For instance, in the feature selection task [22, Section 4.3], the 10-dimensional $Y$ is generated by adding the same source feature $X_1$​ to every dimension of an independent noise vector. This explicitly builds high redundancy into the system (as the information about $X_1$ is repeated across $10$ axes), which SMI favours.
>
> 2. > One small improvement for a reader like myself would be to situate the SMI literature with respect to the neural MI estimation literature...
>
>     Thank you for this suggestion! The primary advantage of SMI is indeed computational simplicity. State-of-the-art neural MI estimators, such as MINDE [r.5], can require several hours for training and estimation. In contrast, estimating SMI is free from any optimization process and can be completed in a minute at most. We have added this clarification to the revised manuscript on lines 68-80.
>
>     [r.5] G. Franzese et al., "MINDE: Mutual Information Neural Diffusion Estimation." Proc. of ICLR 2024
>
> 3. > One other thought that might extend the impact of this work is to discuss whether there are connections to other "sliced" estimators, like sliced score matching. Even the popular Hutchinson trace estimator could be considered a sliced estimator.
>
>     Indeed, we are interested in extending our analysis to other slicing methods. And while the "sliced" Hutchinson estimator proved to be pretty robust (after all, a linear function of a matrix is estimated), we have an intuition that a widely used (760 citations of the original work [r.6]) sliced Wasserstein distance [r.6-7] may also be prone to the same problem.
>
>     [r.6] N. Bonneel et al., "Sliced and Radon Wasserstein barycenters of measures." Journal of Mathematical Imaging and Vision, 2015
>
>     [r.7] S. Kolouri et al., "Generalized Sliced Wasserstein Distances." Proc. of NeurIPS 2019
>
>
> We would like to once again thank Reviewer **AoMn** for carefully reading and reviewing our article. We hope our comments fully address your questions, and we remain ready to address any additional questions you might have.

---

### Author Response · Authors · 2025-11-21
**Revised version uploaded**

Dear Reviewers,

Thank you for taking the time to review our manuscript and for your valuable feedback and suggestions. We were pleased that the overall reception of our work was mostly positive and that our critical assessment of SMI was explicitly appreciated by everyone.

Motivated by your thorough reviews, we have made the following changes to the manuscript:

1. We extended our theoretical analysis beyond the Gaussian case by providing a general result on SMI's saturation in a general continuous, high-redundancy setting — **Lemma C.2 in Appendix C**. While this result is less tight, it demonstrates that saturation is a general phenomenon, not one confined to the Gaussian case (previously, we had only verified this experimentally).
1. Using a simple Gaussian channel experiment (the last paragraph of **Section 6** and **Figure 7**), we provide an explicit example highlighting SMI's inability to detect increases in dependence, even for linear transformations that enhance information extraction. In this setup, $I(X;Y)$ is higher for a $Y$ that yields lower regression error $\mathbb{E} \\|X - \mathbb{E}[X \mid Y]\\|^2$. However, due to its redundancy bias, SMI prefers transformations that correlate features of the channel's input, making $Y$ less suitable for reconstructing $X$.
1. We conducted additional experiments with real data to further support our findings. When applied to real images, SMI both (a) saturates and (b) becomes statistically indistinguishable from zero. For details, please see **Appendix G.3**.
1. **Appendix E** now contains an expanded discussion of our visual abstract, elaborating on our placement of SMI between correlation and MI+copula.
1. We have introduced **Appendix F** to justify our choice for the number of nearest neighbors used in the KSG estimator.
1. Some typos were corrected, and some other minor clarifications were introduced.

All changes are highlighted in red (if an entire new section was introduced, only the corresponding header is colored). We will also address each question and concern individually in our comments below.

Once again, we thank the reviewers for their thoughtful work and insights.

---

### Public Comment · ~Ziv_Goldfeld1 · 2025-11-26
**Contextualizing the limitations of SMI**

The paper raises several important issues about sliced mutual information (SMI). However, many of these points were already surfaced, discussed, and empirically illustrated in prior work. For example:

* The fact that SMI does not satisfy the data processing inequality was explicitly identified and analyzed in Section 3.2 of [20].
* The decay of SMI (and more generally k-SMI) with dimension was quantified and discussed in Section 4.3 of [22].
* Differences between the behavior of SMI and MI in various settings, including InfoMax and InfoGAN, were systematically examined in [20, 22, 44].

In this sense, a number of the paper’s observations are re-statements or extensions of known phenomena, rather than entirely new pathologies.

Based on the initial reviews, it seems that the reviewers were not always closely familiar with SMI, its earlier literature, and its use cases. From that standpoint, the narrative of the current paper can make the situation appear more dramatic than it is, potentially inflating the perceived significance. In reality, SMI was proposed in 2021 as an initial attempt to bypass the statistical curse of dimensionality in MI estimation. It was posed as a surrogate quantity (not a proxy of MI) that inherits some of its properties and is easy to compute and estimate off the bat, warranting further exploration as a useful quantity. It had advantages and disadvantages, many of which were already understood at the time, while others have been explored in subsequent work. This line of research led to max-SMI and optimal-SMI, as well as various heuristic enhancements introduced in applied papers. Indeed, SMI has been used in numerous applications, often with good empirical performance. The current paper raises several interesting points about SMI and some of its less desirable features, but the overall presentation, in my view, paints a more extreme picture than warranted. For instance, a balanced assessment of SMI-based approaches should take the max- and optimal-SMI variants into account, instead of zooming in on the very initial iteration.

Regarding the empirical discussion: the suggestion that previous experiments were somehow contrived to favor SMI seems unfair. In their responses, the authors repeatedly refer to the synthetic construction in [22, Section 4.3], where a 10-dimensional random vector was deliberately designed to probe what types of dependence SMI can and cannot detect. This is a standard stress-test, not a real-data benchmark. In contrast, the real-world experiments in [20, 22, 44] do not impose any special structure, yet SMI-based training performs well in many cases. This does not imply that SMI will always work, but it does suggest that the claim that SMI “usually fails” may be overstated based on the limited number of small-scale examples provided in the paper.

Finally, I find Figure 1 difficult to interpret scientifically. It appears to present a heuristic and subjective idea in a quantitative format, but no objective criterion is provided for determining the “actual” or “perceived” usefulness or ability to capture dependencies. In the absence of such a criterion, the figure seems to reflect the authors’ personal assessment rather than a measurable property of the methods.

---

> ### Author Response · Authors · 2025-11-27
> **Response (1/3)**
>
> Dear Professor Ziv Goldfeld,
>
> Thank you for reviewing our work and for initiating a public discussion on the points that concern you. In this response, we address your questions factually, point by point. For clarity, we have highlighted the key parts in **bold** and apologize if this seems excessive.
>
> 1. > However, many of these points were already surfaced, discussed, and empirically illustrated in prior work.
>
>    We respectfully disagree with you. In this part of your review, you list (a) the violation of DPI, (b) the decay of SMI with growing dimension and (с) differences in behavior between MI and SMI in InfoMax settings as the points that do not require additional academic discussion. We kindly note the following:
>    - (a) and (b) were thoroughly discussed in the Background section in **corresponding paragraphs** (lines 168-182 and 183-193) with **proper attribution and citations** (for (a), **Section 3.2 of [20]** was referred to in **line 174**, and for (b), **Theorem 3 of Section 4.3 of [22]** was cited in **line 188**); previous works on (c) have also been mentioned in **Section 6, lines 345-346**.
>    - In some previous works, (a) is discussed not just as a special property of SMI, but as a useful and interpretable one. For example,
>      >  Unlike MI, SMI can increase with more processing of the variable... This introduces a new perspective on information, where we can interpret SI(X; Y) as the amount of computationally usable information that can be extracted between them, constrained by the computational capacity. [49]
>
>      > Furthermore, in certain aspects, SMI is more compatible with modern machine learning practice than classic MI.  In particular, deterministic transformations of the random variables can increase SMI... [20]
>
>      In our Gaussian channel setup from Section 6, we show that this intuition is not as straightforward as it looks, and DPI violation does not imply that SMI favours "computationally usable information".
>    - In [22], (b) is established only as an asymptotic result (as $d \to \infty$), which is also derived in the Gaussian case. In contrast, we provide a **precise** expression and a **tractable non-asymptotic upper bound** in Lemma 4.1. In addition to more tight results on the decay problem, these precise expressions allow us to discover the **saturation** phenomenon, which **was not discussed** in the previous literature. In lines 192-193, we explicitly state that this is our contribution with regards to (b). We also pinpoint the driving factor behind $\mathsf{SI} \to 0$ as $d \to \infty$ — the **reduncdancy bias**.
>
>      Additionnaly, (b) is experimentally verified for the Gaussian case only in [22]. Our work offers a more diverse set of benchmarks that verify (b) in non-Gaussian settings.
>    - We respectfully disagree that (c) was systematically examined in [20, 22, 44]. These works present *only positive results* on Deep InfoMax tasks, which contradicts our findings. We believe the community should see the full picture; while the previous works consider settings where SMI is effective, we caution that the redundancy bias can lead to collapsed solutions in other settings, such as self-supervised representation learning (see Section 6).
>
> 2. > In this sense, a number of the paper’s observations are re-statements or extensions of known phenomena, rather than entirely new pathologies.
>
>     We never state in our paper that the observations from the Background section are our own. On the contrary, in **lines 151-153** we explicitly state that **some results on (a) and (b) are recapitulated**, and **lines 190-193** highlight that **our results go beyond previous observations**. Furthermore, Section 6 considers InfoMax settings that **were not evaluated in prior work on SMI**.

---

> ### Author Response · Authors · 2025-11-27
> **Response (2/3)**
>
> 3. > From that standpoint, the narrative of the current paper can make the situation appear more dramatic than it is, potentially inflating the perceived significance.
>
>    The Introduction and Background sections of our article are as comprehensive as the page limit allows, providing an overview of SMI's key merits and documented demerits with appropriate attribution. Consequently, we firmly reject the assertion that our work's significance is artificially inflated.
>
> 4. > It was posed as a surrogate quantity (not a proxy of MI)
>
>    Our paper (lines 304-308) explicitly states that SMI is a
>    > **distinct measure** of statistical dependance, and **should not be viewed as an approximation of MI**. Instead, our analysis focuses on the relationship between the two measures...
>
>    We maintain this view throughout the paper and position our findings accordingly.
>
> 5. > It had advantages and disadvantages, many of which were already understood at the time, while others have been explored in subsequent work. This line of research led to max-SMI and optimal-SMI
>
>    We respectfully disagree with this statement. The original works on SMI [20,22] and subsequent works on mSMI [44] and oSMI [17] offer very limited discussion of SMI's limitations. To our knowledge, **neither [20] nor [22] contains a dedicated limitations section**; [20] provides no such discussion, while [22] only addresses the specific issue of $\mathsf{SI}_k \to 0$ without even framing it as a limitation. Furthermore, [44] focuses more on the advantages of mSMI over CCA than its advantages over SMI. Regarding SMI, this work primarily discusses improved convergence rates and better performance in independence testing. Surprisingly, [44] **avoids evaluating SMI on representation learning tasks**. Finally, while [17] provides a motivating example on page 3, it does not detail the specific limitations of SMI that arise from non-optimal slicing.
>
>    **We also kindly note that our Background section already overviews the contribution of these works.**
>
> 6. > the overall presentation, in my view, paints a more extreme picture than warranted.
>
>    We respectfully disagree with Professor Goldfeld. Our work serves to complement the original papers [20, 22], which focus on the positive properties of SMI, such as the nullification property, computational simplicity, and fast convergence rates. We believe it is valuable to have a study that systematically analyzes the limitations of SMI.
>
> 7. > For instance, a balanced assessment of SMI-based approaches should take the max- and optimal-SMI variants into account, instead of zooming in on the very initial iteration.
>
>    In our work, we primarily focus on vanilla SMI for two reasons:
>    - It is the most widely adopted variant. For instance, optimal-SMI is used in only one unpublished work, whereas SMI is applied by many researchers across various fields.
>    - In practice, mSMI and oSMI can be considered restricted variants of MINE (Belghazi et al., 2018). They require neural network optimization, making them significantly more computationally expensive than SMI.
>
>    **However, Section 4.3 still provides a brief discussion of mSMI and oSMI. We also plan to further investigate the limitations of these variants in our future work.**
>
> 8. > the suggestion that previous experiments were somehow contrived to favor SMI seems unfair.
>
>    **This opinion is currently not included in the article** and appears only in our discussion with the reviewers. Motivated by your concerns, we plan to reproduce some of the experiments from [20,22] to further analyze, why SMI behaves better in these setups.
>
> 9. >  In contrast, the real-world experiments in [20, 22, 44] do not impose any special structure
>
>    We kindly ask for an additional time to assess this statement. Despite the experiments featuring real data, we still suspect that they might inherently favor SMI. For example, $X,Y$ from the second experiment of [20, Section 4.3] essentially share the same bit in many pixels, which might be a case of high redundancy. We will further investigate these benchmarks in due course.
>
> 10. > Finally, I find Figure 1 difficult to interpret scientifically... the figure seems to reflect the authors’ personal assessment rather than a measurable property of the methods.
>
>     This is indeed true and will be stated explicitly in the paper.

---

> ### Author Response · Authors · 2025-11-27
> **Response (3/3)**
>
> **Summary:**
>
> We thank Professor Goldfeld for his time and attention to our work! However, we strongly disagree with the claim that our paper merely recapitulates previous results: the Introduction and Background sections comprise only three pages, while approximately 18 pages are dedicated to our original contributions. We also maintain our view that the community deserves a work that analyzes and explains deficiencies of SMI, as [20,22,44] do not provide an adequate discussion of them.
>
> Compared to the existing literature, we:
> - Provide a detailed analysis of the $\mathsf{SI} \to 0$ problem, including precise closed-form expressions, extensive non-Gaussian experiments, and an explanation for this phenomenon.
> - Demonstrate that $\mathsf{SI} \to 0$ is a significant issue (see Section G.3).
> - Identify **brand new** deficiencies of SMI: **saturation** and **redundancy bias**.
> - Show that SMI leads to collapsed representations in SSRL, even when collapse-mitigation strategies are applied.
> - Show that SMI does not necessarily prefer linear transformations that improve information extraction.
>
> Furthermore, our work makes a more explicit point that the dynamics of SMI typically do not align with those of MI in low-redundancy settings, which is a crucial point for applications that has been overlooked in the literature.

---

> > ### Comment · Area_Chair_gWDq · 2025-11-27
> > **Saturation**
> >
> > Thanks for replying. You've raised "discovering the saturation phenomena" as a contribution of your work. However, isn't this an immediate consequence of Proposition 1, part 2 of the original SMI paper?

---

> > ### Comment · Area_Chair_gWDq · 2025-11-27
> > **Redundancy**
> >
> > Similarly, note that while "redundancy" was not explicitly called out there by name, it can be immediately seen by reading Proposition 4 in the original SMI paper - this is indeed why the second part of the proposition requires a lower bound on the singular values.

---

> ### Comment · Area_Chair_gWDq · 2025-11-27
>
> I will also point out that even the title of your paper calls SMI "deceptive" -  throughout your paper, language and discussion repeatedly and strongly pushes this notion that SMI is "flawed".

---

> ### Author Response · Authors · 2025-11-27
>
> Dear AC,
>
> Thank you for your continued engagement. As suggested, we have removed the mention of a potential conflict of interest. We are also glad to be assured that the other reviewers are familiar with Professor Goldfeld's contributions.
>
> For the most efficient and organized discussion, we would be grateful if you could consolidate your questions into a single message. This helps ensure our responses are comprehensive and don't get overlooked, and we will happily reciprocate.
>
> Below are our responses:
>
> > Public comments weighing in to the decision on a paper at ICLR is completely appropriate. What matters is the facts, and it is inappropriate to indicate that any comment should be dismissed no matter the source.
>
> We fully agree with the AC. However, the AC's statement that Prof. Goldfeld's comment "would support rejection of this work," (original revision of the comment) despite the paper's positive reception by the reviewers, put us on high alert.
>
> > You've raised "discovering the saturation phenomena" as a contribution of your work. However, isn't this an immediate consequence of Proposition 1, part 2 of the original SMI paper?
>
> No. Our result is much stronger than Proposition 1, part 2 of [20]. For instance, given the example from Lemma 4.1, Prop. 1 p. 2 [20] yields $\mathsf{SI}(X;Y) \leq \frac{1}{d} \mathsf{I}(X;Y)$, while our result suggests $\mathsf{SI}(X;Y) \leq \frac{1}{d-1}$, **which does not depend on mutual information.** Therefore, the saturation is strong (even $\mathsf{I}(X;Y) \to \infty$ does not break it) and can not be explained solely by non-optimality of projections.
>
> We also note that our bound is much harder to derive compared to Prop. 1 p. 2 [20]: while the latter is almost obvious, the former is more involved.
>
> > note that while "redundancy" was not explicitly called out there by name, it can be immediately seen by reading Proposition 4
>
> We kindly ask the AC to clarify this point. Our results indicate that SMI prefers joint distributions $\mathbb{P}_{X,Y}$ that are degenerate (supported on a low-dimensional linear manifold). This behavior also extends to nonlinear transformations of $X$ and $Y$, as illustrated in Section 6. In contrast, Proposition 4 of [20] suggests that SMI is indifferent to being maximized over non-injective linear mappings, whether they act within the same space (via square matrices $A$ of rank $r < d$) or between different spaces (via non-square matrices $B$).
>
> > I will also point out that even the title of your paper calls SMI "deceptive"... If you are merely contending that there are limitations to SMI, your paper would need to be drastically revised, no?
>
> We understand this concern. However, we believe that our position is well justified within the text and our response to Professor Goldfeld:
>
> > (**lines 73-79**) Despite its popularity, the research community has largely overlooked potential shortcomings of SMI. Some studies prematurely attribute their results to underlying phenomena without rigorously investigating whether they stem from artifacts introduced by random projections. Furthermore, existing works fail to comprehensively address issues related to random slicing, focusing primarily on suboptimality of random projections for information preservation [17, 44]
>
> > Our work serves to complement the original papers [20, 22], which focus on the positive properties of SMI, such as the nullification property, computational simplicity, and fast convergence rates. We believe it is valuable to have a study that systematically analyzes the limitations of SMI.
>
> Since many studies completely ignore even well-established shortcomings of SMI (such as those noted by Professor Goldfeld), and some even interpret them (like DPI violation) as merits [46,47,48,49], we contend that a separate, dedicated paper is necessary to thoroughly address the limitations of SMI and to caution practitioners about its non-trivial, *deceptive* dynamics. Our aim is not to assert that SMI is "flawed" beyond redemption, but to highlight that many of its critical issues have been overlooked or disregarded.
>
> Throughout our paper, we have maintained a respectful and academic tone. We would be glad to revise specific instances where our language may fall short if the AC indicate them.

---

> ### Comment · Area_Chair_gWDq · 2025-11-28
> **Answering questions**
>
> To address your questions, consider the following:
>
> * "For instance, given the example from Lemma 4.1, Prop. 1 p. 2 [20] yields $\mathsf{SI}(X;Y) \leq \frac{1}{d} \mathsf{I}(X;Y)$, while our result suggests $\mathsf{SI}(X;Y) \leq \frac{1}{d-1}$, which does not depend on mutual information."
>
> It is true that this is tighter, recalling that Prop 1 [20] is the general case, while your Lemma 4.1 is a special case. Note, however, that the discussion of Prop 1 [20], specifically Remark 4 and footnote 3, also pointed out this behavior (finite SMI, infinite MI) in a fairly related Gaussian example to the one you use. Your discussion of your example is laudably much clearer and more full, but all the important points seem to me to be present in [20] already.
>
> * Information redundancy and Prop 4 in [20]
>
> While I'm not fully certain I understand the precise definition of what you mean by saying SMI prefers information redundancy, I believe you mean that maximizing SMI will encourage repetitions of the same strong k-dimensional informative signal across many directions, rather than encouraging new sources of information. This seems to be the exact point of Prop 4 in [20]. Indeed, increasing diverse information seems to be a reason to choose larger k in k-SMI, rather than limiting to 1-SMI.
>
> * (Potential) misuse of SMI in the literature.
>
> Unfortunately, I think this is the crux of why I don't think your present paper is complete as-is. Demonstrating large-scale misuse and misunderstanding of SMI is critical for establishing that clarifying these limitations is sufficient for a paper. Yet beyond a few short sentences referring to a line of work from a single set of authors and a single paper on privacy, I don't see discussion that really demonstrates that this misunderstanding either occurred or is widespread. Can you be more specific and detailed regarding issues you have with these works that use SMI?

---

> > ### Author Response · Authors · 2025-11-28
> > **Response to AC (1/3)**
> >
> > Dear AC,
> >
> > Thank you for the clarifications. We value an honest review of our work — this is the whole purpose of OpenReview. We never aimed to offend anyone with our paper, only to highlight important problems.
> >
> > > It is true that this is tighter, recalling that Prop 1 [20] is the general case, while your Lemma 4.1 is a special case.
> >
> > We also provide a more general (non-Gaussian) result in Lemma C.2 of Appendix C. This bound is also independent of $\mathsf{I}(X;Y)$ and is therefore tighter than that in Prop. 1 of [20]. We also wish to stress the core difference between our results and part 2 of Prop. 1 in [20]: their bound is a direct consequence of the sub-optimality of random projections, whereas ours demonstrates that saturation is a more involved phenomenon that cannot be explained merely by a non-optimal choice of $\phi$ and $\theta$.
> >
> > > Note, however, that the discussion of Prop 1 [20], specifically Remark 4 and footnote 3, also pointed out this behavior
> >
> > We kindly note that Remark 4 in [20] only states that "the gap between MI and SMI may not be bounded" (which does not imply saturation; consider $\mathsf{SI}(X;Y) = \frac{1}{2} \mathsf{I}(X;Y)$). It also does not position this as a fundamental problem, whereas our work explicitly identifies **saturation** as a flaw that **makes SMI unusable in high-MI regimes**.
> >
> > Footnote 3 is closer to our example but relies on a distribution with undefined MI, as $\mathbb{P}_{X,Y}$ is not absolutely continuous with respect to $\mathbb{P}_X \otimes \mathbb{P}_Y$, making Eq. 1 of [20] undefined.
> >
> > Nevertheless, following our discussion with the AC, we acknowledge the importance of Footnote 3 in Remark 4 of [20] and will add a corresponding reference and discussion in our next revision.
> >
> > > While I'm not fully certain I understand the precise definition of what you mean by saying SMI prefers information redundancy
> >
> > It can be formalized as extremely low or even undefined ("$-\infty$") differential entropy of $X$ and $Y$ with relatively high differential entropy of individual components $X_i$ and $Y_i$.
> >
> > > I believe you mean that maximizing SMI will encourage repetitions of the same strong k-dimensional informative signal across many directions, rather than encouraging new sources of information.
> >
> > This is indeed a direct consequence of the redundancy bias, as observed in Section 6.
> >
> > > This seems to be the exact point of Prop 4 in [20].
> >
> > We are still unsure how Prop 4 [20] indicates the redundancy bias. It already considers non-injective (non-full-rank) transforms, which already impose high-redundancy structure by mapping $X$ and $Y$ to manifolds of lower dimensionality. We do not think that this Proposition highlights any flaws of SMI.
> >
> > > Indeed, increasing diverse information seems to be a reason to choose larger k in k-SMI, rather than limiting to 1-SMI.
> >
> > We agree that increasing $k$ is a natural way to reduce redundancy bias. However, our results suggest that only the extreme case of $k=d$ fully alleviates this problem. $k$-SMI still prefers degenerate solutions that lie on $k$-dimensional linear manifolds, which is further supported by the additional experiments we are currently conducting in response to Professor Goldfeld's concerns.
> >
> > > I don't see discussion that really demonstrates that this misunderstanding either occurred or is widespread.
> >
> > For the brevity of the discussion, we decided to not reiterate every concern we raised in **Impact** paragraph of Section 7, which considers a more diverse set of problems.

---

> ### Author Response · Authors · 2025-11-28
> **Response to AC (2/3)**
>
> > Can you be more specific and detailed regarding issues you have with these works that use SMI?
>
> Our issues with the previous works are as follows:
> - [20,22] ("unsurprisingly", as stated by Reviewer ZesV) portray a very positive view of SMI, lacking any dedicated sections on its limitations. Discussions of SMI's potential demerits are fragmented and scattered, and these drawbacks are never framed as issues to begin with.
>
>    For example, [20] mentions that "the gap between MI and SMI may not be bounded" but immediately dismisses this by framing it merely as an example of SMI not being a proxy for MI, which is not a problem in itself, unlike saturation. Similarly, [22] acknowledges that SMI might decay with increasing dimensionality, but in the very next sentence, it employs the "blessing of dimensionality" (Remark 4 [22]), as if to sugar this pill.
>
> - [17,44] provide a very limited critical assessment of SMI (please, refer to our previous discussion and Background section). We were also surprised to see that [44] **explicitly avoided evaluating SMI in SSRL** for no concrete reason — a gap we fill in Section 6.
>
> - [20,22,44] provide only positive results for InfoMax-like tasks, whereas our experiments suggest that there are natural settings where SMI maximization leads to a complete collapse.
>
> - [20,22,37] propose using SMI for independence testing, but do not discuss how $\mathsf{SI} \to 0$ can affect such usage.
>
> - After revisiting the list of works that cite [20,22], we can also append to our list the following:
>    1. (Chen et al. 2022) employ SMI as an optimization objective and seem to be unaware even of the problems mentioned by Professor Goldfeld.
>    2. (Chen et al. 2023) later utilize the same slicing technique for learning a sufficient statistic. Their proof of Theorem 1 in Appendix A relies on $\text{argmax } \mathsf{I} = \text{argmax } \mathsf{SI}$, which is used **without a formal proof**: "It is then easy to verify $\text{argmax } \mathsf{I} = \text{argmax } \mathsf{SI}$". Our result from Section 6 indicates that this is generally not the case. Therefore, **we are absolutely certain that a more explicit warning on SMI for InfoMax should be provided in the literature.**
>    3. (He et al. 2024) employ SMI and provide a brief overview of its merits, but mention only one "limitation": the fact that SMI < MI, which is not a limitation at all.
>    4. (Dentan et al. 2025) leverage pointwise SMI to predict memorization in NNs. While the results look promising, the work lack a proper discussion of SMI (even the merits). PSMI provides a decent ROC curve, but one should be very aware before comparing the results between different models, as changing the dimensionality and redundancy of information affects SMI significantly. As we said, this possible limitation is not addressed in (Dentan et al. 2025).
>    5. (Shaeri & Middel, 2025) use SMI to highlight that their method "prioritizes the most informative neurons." Our work indicates that **SMI might not measure the information content, but the redundant information**. Therefore, this conclusion should be cautiously reassessed.
>    6. (Xiao et al., 2024) construct attention maps via measuring SMI in latent spaces of vision models, introducing this measure with little to no discussion (even in regard to the merits). While we are currently not able to review this work due to difficult notation, we again emphasize that SMI might not measure the true dependencies in high-dimensional regimes, and should be used cautiously.
>
> Y. Chen et al., "Scalable Infomin Learning." Proc. of NeurIPS 2022.
>
> Y. Chen et al., "Is Learning Summary Statistics Necessary for Likelihood-free Inference?" Proc. of ICML 2023.
>
> Z. Hu et al., "nfoNet: Neural Estimation of Mutual Information without Test-Time Optimization." 2024, arXiv:2402.10158
>
> J. Dentan et al., "Predicting Memorization Within Large Language Models Fine-Tuned for Classification." Proc. of ECAI 2025"
>
> P. Shaeri and A. Middel, "MID-L: Matrix-Interpolated Dropout Layer with Layer-wise Neuron Selection." 2025, arXiv:2505.11416
>
> K. Xiao et al. "Sliced Maximal Information Coefficient: A Training-Free Approach for Image Quality Assessment Enhancement." Proc. of ICME 2024.
>
> In total, we discuss 13 problematic works: 7 in our manuscript and 6 in this reply. For 11 of these, we provide a concrete explanation of our concerns, ranging from framing the DPI violation as a merit to assuming $\text{argmax } \mathsf{I} = \text{argmax } \mathsf{SI}$ without proof. For the remaining two, we acknowledge the role of SMI in their contributions but caution against potential problems when comparing SMI values across different experimental setups without proper knowledge of SMI's biases.

---

> ### Author Response · Authors · 2025-11-28
> **Response to AC (3/3)**
>
> In contrast to the previous works, and in addition to the contributions that were already mentioned during this discussion, we
> - Are more vocal about the demerits of SMI (**while also restating the key merits and referring the reader to [20,22] for additional information on useful properties of SMI in Background section**);
> - Explicitly identify $\mathsf{SI} \to 0$ as a **serious problem** for two key reasons
>    1. It renders SMI estimates across different dimensionalities **fundamentally incomparable**, as they are theoretically bounded by factors depending on $d$ (lines 242-244).
>    2. In practical high-dimensional scenarios, SMI becomes indistinguishable from zero, even when the theoretical "blessing of dimensionality" holds. This occurs even for variables with non-zero classic MI, as demonstrated in Section G.3, where SMI between independent vectors is equivalent to that of dependent vectors with $\mathsf{I}(X;Y) = 0.5 \text{ nat}$.
> - Explicitly state that the DPI violation warrants further academic discussion **(lines 87-90)**, as it is not a direct indicator that SMI is "computationally usable information" [49] or "more compatible with modern machine learning practice than classic MI" [20].
> - Provide explanations of the deficiencies of SMI and investigate how they can affect practical results (e.g., lead to collapsed solutions, as in Section 6).

---

### Comment · Area_Chair_gWDq · 2025-11-27
**To authors and Reviewers**

Authors and reviewers for "Curse of Slicing: Why Sliced Mutual Information is a Deceptive Measure of Statistical Dependence". Note that a public comment has been posted that raises severe issues for this submission that in my assessment could support rejection of this work. Please read this comment and respond to it and adjust your assessment and reviews as appropriate.

---

> ### Comment · Reviewer_ZesV · 2025-11-27
> **Considering public comment**
>
> I have looked back over references [20,22,44], and considered how they are presented in the submitted paper.
> I appreciate the public comment, but I do not think it changes my score, and I disagree with the Area Chair that this comment raises "severe issues" or suggests rejection.  If anything, it encourages me to recommend acceptance.
>
> The comment raises that previous analysis in [20,22,44] cover issues covered in this paper.  First, I do not think this paper suggests otherwise;  they reference these papers and describe their contents in a way that seems acceptable.  As an example, the public comment states that "The fact that SMI does not satisfy the data processing inequality was explicitly identified and analyzed in Section 3.2 of [20]." Indeed, the submission clearly states that the fact that SMI does not satisfy DPI appears in [20].  (If an author from those papers wants to argue they could or should have presented differently, they're certainly entitled to that opinion, but I did not see anything that I would consider inappropriate in how those papers were discussed in this submission, including from the discussion in the public comment.)  Second, the papers [20,22,44] clearly (and unsurprisingly) take a much more positive view towards the benefits of SMI and its variants. Having a paper offer a mathematically grounded counterpoint seems appropriately useful in this context.
>
> In both the submission and in the comments here I do not thing the authors have made excessive comments regarding the issues they perceive with SMI.  I continue to think this submission provides a useful counterpoint to the previously published work, and would continue to support acceptance.

---

> > ### Comment · Area_Chair_gWDq · 2025-11-27
> >
> > To clarify my concerns - I believe this paper is incorrectly framed, misrepresents the status of the literature, and doesn't provide new information beyond clarifying existing results.

---

> ### Author Response · Authors · 2025-11-27
>
> Dear Area Chair gWDq,
>
> We understand the concerns raised in the public comment and are currently preparing a point-by-point response. We will provide our clarifications as soon as possible.
>
> We would also like to declare a potential conflict of interest. We were contacted by the first author of the SMI and $k$-SMI papers, and the second author of the max-SMI paper, Ziv Goldfeld. While we greatly respect Professor Goldfeld and his contributions to the field, and are pleased to have been contacted, we were unsure of the appropriate procedure since this connection was not stated in his own comment. Our solution is to make this information known to you.

---

### Author Response · Authors · 2025-12-03
**General response and revision**

Following the ICLR 2026 Program Chairs' message regarding the reassignment of area chairs, we summarize the progress of the rebuttal.

We thank the reviewers for their insightful comments and are pleased that our responses addressed their concerns, as reflected in the updated scores (**8 8 8 6**).

Subsequently, the Area Chair invited Prof. Goldfeld, an author of the original SMI, k‑SMI and max‑SMI papers, to provide his expert opinion. We then addressed the points raised by Prof. Goldfeld and the Area Chair in the public comment "Contextualizing the limitations of SMI", offering detailed, point‑by‑point responses. Following that exchange, reviewer **ZesV** noted that no significant issues has been raised and reaffirmed support for acceptance, viewing our paper as a useful counterpoint to the existing literature.

We are grateful to all reviewers, the Area Chair, and Prof. Goldfeld for their feedback, which has strengthened the manuscript. In response to the discussion, we have made the following additions and improvements in the final revision (newly introduced revisions as well as previously made changes are highlighted in red):

1. We revisited the feature extraction and independence testing experiments from the original SMI papers [20,22] (see Section 7 & Appendix I).
    - For feature extraction, we reproduced the Gaussian setup of [20], but increased the number of relevant features. Both visualizations of the learned feature matrix and a quantitative assessment via the effective rank **confirm the redundancy bias of SMI** identified in our work.
    - For independence testing, we replicated the protocol of [20] with one practically motivated modification. Namely, we pool SMI (and MI) estimates across multiple dimensions for each sample size before computing the ROC‑AUC. This reflects a more realistic scenario where a single threshold must be chosen a priori. Under this setup, **SMI proves less reliable than MI on most of our benchmarks, contrasting with the original findings**.
2. We have improved our non‑Gaussian analysis by providing a more general result on the decay and saturation of $k$-SMI (**Lemma C.2**).
3. To underscore the importance of our findings, **we have examined 13 prior works** that we consider problematic (7 of which were present in the manuscript from the beginning, and additional 6 were provided in our response to Prof. Goldfeld). For 11 of these, we detail specific concerns, ranging from incorrectly framing the violation of the Data Processing Inequality as a merit to relying on unsupported assumptions about SMI maximization. For the remaining two, we acknowledge SMI's role, but caution against comparing SMI values across different experimental setups without accounting for its inherent biases. The **Impact paragraph** has been updated accordingly.
4. In line with suggestions from the Prof. Goldfeld and Area Chair, we have made several minor adjustments (e.g., clarified that the visual abstract in Figure 1 represents our informed perspective, etc.).

Should any further questions arise, we invite the readers to consult our detailed, point-by-point replies in the public comment history, where all concerns have been thoroughly addressed.

*Note*: The addition of new references in the updated Impact section has altered the citation numbers in the manuscript. For consistency with all previous replies, we retain the original numbering in this comment.

---

### Meta-Review · Area_Chair_FquC · 2026-01-04

**Summary:**

Reviewers raised several concerns including the significance of the contribution, the validity of the assumptions used in the theoretical analysis, and the sufficiency of the empirical evaluation. These issues have been addressed in the rebuttal and the revised manuscript, and all reviewers are satisfied and support acceptance. I have also carefully read the paper and agree with the reviewers’ assessments. Therefore, I recommend acceptance of the paper.

**Reviewer Concerns:**

I believe that no significant concerns remain after the rebuttal.

**Reviewer Scores:**

**Reviewer AoMn** raised a concern regarding the significance of the results, which has been addressed in the rebuttal. Therefore, I believe this reviewer will maintain the original positive score.

**Reviewer KQcg** raised concerns about the strong assumptions used in the theoretical analysis and the lack of empirical evaluation in cases where the assumption does not hold. The authors carefully addressed these issues, and the reviewer will increase the score.

**Reviewer ZesV** did not raise any significant concerns and is therefore expected to maintain the original positive score.

**Reviewer pnxw** pointed out insufficient empirical evaluation, unclear presentation of the experimental protocol, and raised several questions, including the usecase of SMI instead of MI. These issues were addressed in the rebuttal, and the reviewer is willing to increase the score.

---

### Decision · Program_Chairs · 2026-01-26

Accept (Poster)